# The recovery of standing and locomotion after spinal cord injury does not require task-specific training

Jonathan Harnie[1], Adam Doelman[1], Emmanuelle de Vette[1], Johannie Audet[1], Etienne Desrochers[1], Nathaly Gaudreault[2], Alain Frigon[1]*

[1]Department of Pharmacology-Physiology, Faculty of Medicine and Health Sciences, Université de Sherbrooke, Centre de Recherche du CHUS, Sherbrooke, Canada; [2]School of Rehabilitation, Faculty of Medicine and Health Sciences, Université de Sherbrooke, Centre de Recherche du CHUS, Sherbrooke, Canada

**Abstract** After complete spinal cord injury, mammals, including mice, rats and cats, recover hindlimb locomotion with treadmill training. The premise is that sensory cues consistent with locomotion reorganize spinal sensorimotor circuits. Here, we show that hindlimb standing and locomotion recover after spinal transection in cats without task-specific training. Spinal-transected cats recovered full weight bearing standing and locomotion after five weeks of rhythmic manual stimulation of triceps surae muscles (non-specific training) and without any intervention. Moreover, cats modulated locomotor speed and performed split-belt locomotion six weeks after spinal transection, functions that were not trained or tested in the weeks prior. This indicates that spinal networks controlling standing and locomotion and their interactions with sensory feedback from the limbs remain largely intact after complete spinal cord injury. We conclude that standing and locomotor recovery is due to the return of neuronal excitability within spinal sensorimotor circuits that do not require task-specific activity-dependent plasticity.

*For correspondence:
Alain.Frigon@USherbrooke.ca

**Competing interests:** The authors declare that no competing interests exist.

## Introduction

Transection of the spinal cord at thoracic levels completely and permanently abolishes communication between the brain and motoneurons located at lumbar levels that activate leg muscles. Despite this complete disruption, studies have shown that hindlimb locomotion recovers in various pre-clinical models, such as cats, rats and mice, with treadmill locomotor training (*Shurrager and Dykman, 1951*; *Lovely et al., 1986*; *Barbeau and Rossignol, 1987*; *Lovely et al., 1990*; *Hodgson et al., 1994*; *Bélanger et al., 1996*; *De Leon et al., 1998b*; *De Leon et al., 1999a*; *Leblond et al., 2003*; *Cha et al., 2007*; *Sławińska et al., 2012*). This remarkable recovery is due to the presence of a network of neurons located within the lumbar cord that produces the basic locomotor pattern, termed the central pattern generator (**CPG**) (*Grillner, 1981*; *McCrea and Rybak, 2008*; *Rossignol and Frigon, 2011*; *Kiehn, 2016*). Treadmill training, which consists of reproducing a pattern consistent with locomotion, assisted manually by therapists or with a robotic device, is thought to provide sensory cues from muscles, joints and skin that essentially teach the spinal CPG to function without signals from the brain (*Harkema, 2001*; *Dietz and Harkema, 2004*; *Edgerton et al., 2008*; *Edgerton and Roy, 2009*; *Brownstone et al., 2015*). Thus, the beneficial effects of treadmill training are based on the premise that task-specific mass practice of walking reorganizes spinal sensorimotor circuits, which learn to generate a locomotor pattern once again.

Consistent with the premise that locomotor training must be task specific is the demonstration that spinal-transected cats trained to stand maintained standing for longer periods compared to treadmill-trained cats but had difficulty generating hindlimb locomotion (*Hodgson et al., 1994*). In

contrast, spinal-transected cats trained to step on a treadmill generated robust hindlimb locomotion but could not maintain standing for prolonged periods. However, is comparing standing and loco-motor training adequate to demonstrate that training must be task specific? During stand training, sensory feedback from the legs is tonic, whereas during treadmill training it is phasic. If sensory cues from the periphery reorganize spinal sensorimotor circuits, two types of phasic sensory inputs, with one being non-task-specific, would provide a better comparison to demonstrate the principle of task-specificity for training. Another caveat is that most studies that have demonstrated benefits of treadmill locomotor training in pre-clinical models did not include a control non-trained group (*Lovely et al., 1990*; *De Leon et al., 1999a*; *De Leon et al., 1999b*), or if they did, the recovery in these animals was largely ignored or attributed to self-training (*Lovely et al., 1986*; *Hodgson et al., 1994*; *De Leon et al., 1998a*; *De Leon et al., 1998b*). As such, the benefits of treadmill training in animal models of spinal transection is, at best, unclear. Despite a lack of clear evidence that task-specific locomotor training is required for the recovery of hindlimb locomotion in animal models, body-weight supported treadmill training (BWSTT), without or with robotic devices, was adopted in people with spinal cord injury (SCI) and became the gold standard for restoring walking (*Visintin and Barbeau, 1989*; *Fung et al., 1990*; *Wernig and Müller, 1992*; *Visintin and Barbeau, 1994*; *Wernig et al., 1995*; *Harkema et al., 1997*; *Barbeau et al., 1998*; *Behrman and Harkema, 2000*; *Colombo et al., 2000*; *Dobkin et al., 2006*). However, the lack of appropriate controls or significant differences with overground gait training and other forms of physiotherapy in human studies has led to questions regarding the benefits of BWSTT (*Wolpaw, 2006*; *Dobkin et al., 2006*; *Dobkin and Duncan, 2012*; *Mehrholz et al., 2012*; *Mehrholz et al., 2017*).

Therefore, the main goal of this study was to determine if the recovery of hindlimb locomotion after complete SCI in adult cats requires task-specific training. To address this goal, we compared two types of training that provide phasic sensory inputs to spinal circuits, one task-specific (locomotor training) and another non-task-specific (manual therapy). We also included a third group that received no intervention to determine if locomotor recovery occurs spontaneously. After spinal transection, we also tested weight bearing during standing to determine if this function recovered without task-specific training. We hypothesized that the recovery of standing and hindlimb locomotion does not require task-specific training after spinal transection.

## Results

The main goal of the present study was to determine if the recovery of standing and hindlimb locomotion after complete SCI in the cat model requires task-specific training. To accomplish this goal, we compared the recovery of weight bearing during standing and hindlimb locomotion in spinal-transected cats that received non-task-specific training in the form of rhythmic manual stimulation of the triceps surae muscles (Group 1, Non-specific), locomotor training (Group 2, Locomotor-trained) and no training (Group 3, Untrained). We evaluated the recovery of standing and hindlimb locomotion each week after spinal transection in all but three cats (Cat 1 and Cat 2 from Group 1 and Cat 12 from Group 3). Chronologically, these were the first three cats experimented upon to avoid a training effect from testing. However, as we observed substantial recovery in these cats, we decided to perform weekly testing for subsequent animals to determine the time course of recovery (*Figure 1A*). During weekly testing in the first 5 weeks after spinal transection, the ability to stand without and with perineal stimulation was tested for ~15 s in each condition, whereas hindlimb locomotion was tested at a treadmill speed of 0.4 m/s without and with perineal stimulation for ~30 s in each condition. At week six after spinal transection, we did not test standing, but we performed a more thorough investigation of locomotor performance, as described below.

During the spinal transection surgery, we ensured that the lesion was complete by opening and cleaning a gap of ~0.5 cm between the rostral and caudal cut ends. As the spinal cord is naturally under tension, it retracts rostrally and caudally when cut. A histological analysis confirmed that the spinal transection was complete in all cats (*Figure 1C*). The following results show that the recovery of standing and hindlimb locomotion after complete SCI in the adult cat does not require task-specific training. We also show that the ability to modulate speed and to step on a split-belt treadmill does not require task-specific training. Instead, we show that the recovery depends on the return of excitability within spinal sensorimotor circuits.

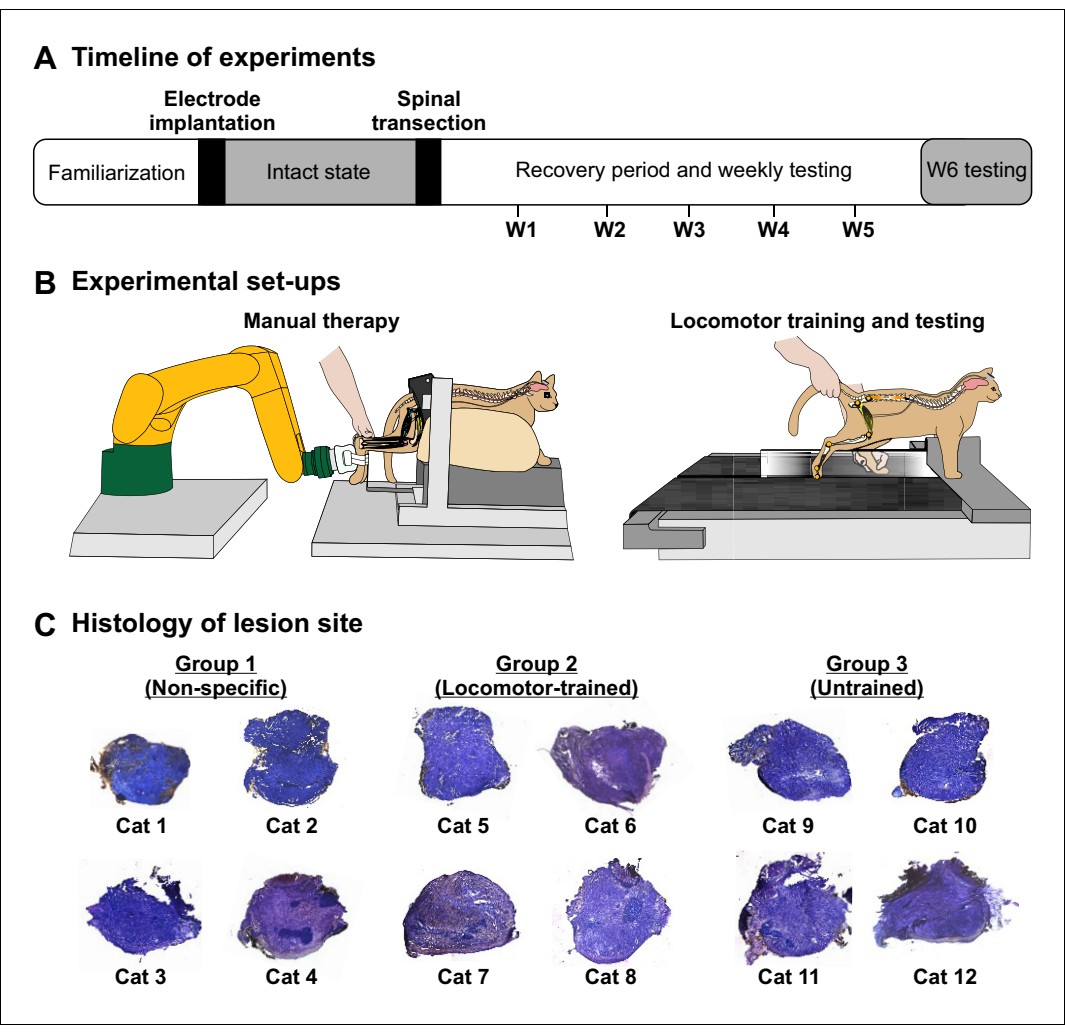

**Figure 1.** Schematic representation of experimental timeline and set-ups. (**A**) Timeline of experiments. After transection, we performed stand and locomotor testing each week (W1–W5) in 9 of 12 cats and locomotor testing in all cats at W6. (**B**) Experimental set-ups for the application of manual therapy (left panel) and for locomotor testing/training (right panel). (**C**) Histological analysis of spinal lesion site.

## Weight bearing during standing recovers spontaneously after spinal transection

We quantified the recovery of weight bearing during standing without and with perineal stimulation for the first 5 weeks after spinal transection for ~15 s in each condition in nine individual cats from the three experimental groups, as three cats were not tested weekly. We stimulated the perineal region by manually pinching the skin with the index and thumb, which provides a general increase in spinal neuronal excitability in spinal-transected animals through an undefined mechanism (*Rossignol et al., 2006*; *Alluin et al., 2015*). We rated weight-bearing recovery on a 6-point scale with 0 representing no weight bearing and five representing full

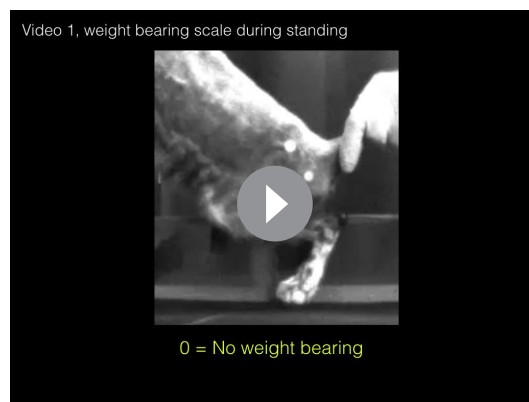

**Video 1.** Standing performance scale. The video shows the rating of standing performance on a six-point scale, from 0 to 5.
https://elifesciences.org/articles/50134#video1

weight bearing with stomping in place, which occurred with perineal stimulation (see Methods, *Video 1*). Seven of nine cats, including at least two animals from each group recovered full weight bearing during standing, a value of 4, without perineal stimulation 1–2 weeks after spinal transection (*Figure 2A*, left panels). *Video 2* shows a cat from each group two weeks after spinal transection performing standing with a value of 4, representing full weight bearing, without perineal stimulation. In Group 2, two cats did not recover weight bearing during standing without perineal stimulation at week five after spinal transection (an example is shown in *Video 3*). With perineal stimulation, we observed full weight bearing during standing in all nine cats four weeks post-transection and some stomping, a value of 5, as early as week one in Cat 4 (*Figure 2A*, right panels).

The effect of perineal stimulation on weight bearing during standing varied between cats, which did not depend on the group they belonged to (*Figure 2B*). For instance, Cat 7, from the Locomotor-trained group, did not display weight bearing during standing without perineal stimulation at week 5 (a value of 0) and weak activity in the ankle extensor soleus, mostly on the right side (*Figure 2B*, top panel; *Video 4A*). However, with perineal stimulation, we observed strong tonic activity in the soleus bilaterally with perineal stimulation and full weight bearing during standing (a value of 4). In another cat, Cat 6, also from the Locomotor-trained group, we observed bilateral activity in soleus and full weight bearing during standing without perineal stimulation at week 5, which was not visibly affected by perineal stimulation, although we did observe a slight elevation of the pelvis (*Figure 2B*, middle panel; *Video 4B*). In Cat 10, from the Untrained group, who also displayed tonic soleus activity bilaterally and full weight bearing during standing without perineal stimulation, perineal stimulation generated stomping, characterized by alternating bursts of activity in the soleus and tibialis anterior bilaterally at week 5 (*Figure 2B*, bottom panel; *Video 4C*). Across cats, perineal stimulation significantly increased the mean EMG amplitude of the soleus muscle during weight bearing at each of the 5 weeks, without significantly affecting tibialis anterior activity (*Figure 2C*, paired t-tests).

The observation that weight bearing during standing recovered in animals of all three groups without and with stimulation of the perineal skin indicates that it occurred spontaneously, without the need for targeted stand training. The different patterns of responses from one animal to another also indicates that, despite a similar SCI, the excitability of spinal neuronal circuits controlling weight support during standing and how these interact with inputs from the skin are variable between animals, independent of the intervention they received.

## Locomotor recovery does not require task-specific training after spinal transection

As stated in the introduction, locomotor training after SCI is thought to provide sensory cues that teach the spinal locomotor CPG to function without signals from supraspinal structures. To determine if these sensory cues need to be task-specific, we assessed locomotor recovery after complete SCI in three experimental groups (Group 1: Non-specific, Group 2: Locomotor-trained and Group 3: Untrained). Six weeks after spinal transection, seven and eight cats out of twelve without and with perineal stimulation, respectively, stepped at the highest possible value of 8 on our locomotor performance scale, including at least two cats from each group (*Figure 3A*). In other words, these cats could step at a range of different speeds during tied-belt and split-belt treadmill locomotion with out-of-phase left-right alternation, proper digitigrade paw placement and full weight bearing. *Video 5* shows examples from three cats, one from each group, stepping at 0.4 m/s without perineal stimulation at week six after spinal transection. Notably, this was the first testing session for Cat 1 (Non-specific) and Cat 12 (Untrained) that were not tested weekly after spinal transection.

*Table 1* breaks down the locomotor performance of individual cats six weeks after spinal transection. Four cats could not perform hindlimb locomotion six weeks after spinal transection without perineal stimulation and two of these cats could not step with perineal stimulation (*Figure 3B*, gray areas). In these cats, the hindlimbs simply dragged behind the body or performed short uncoordinated steps without weight bearing (*Video 6*). Interestingly, the two cats (Cats 7 and 8) that did not express hindlimb locomotion after spinal transection were from the Locomotor-trained group. This is not entirely surprising because these cats also did not recover weight bearing without perineal stimulation (*Figure 2A*). We found no significant difference in the locomotor scores between groups at 6 weeks post-transection without (p=0.52, one-way ANOVA) and with (p=0.38, one-way ANOVA)

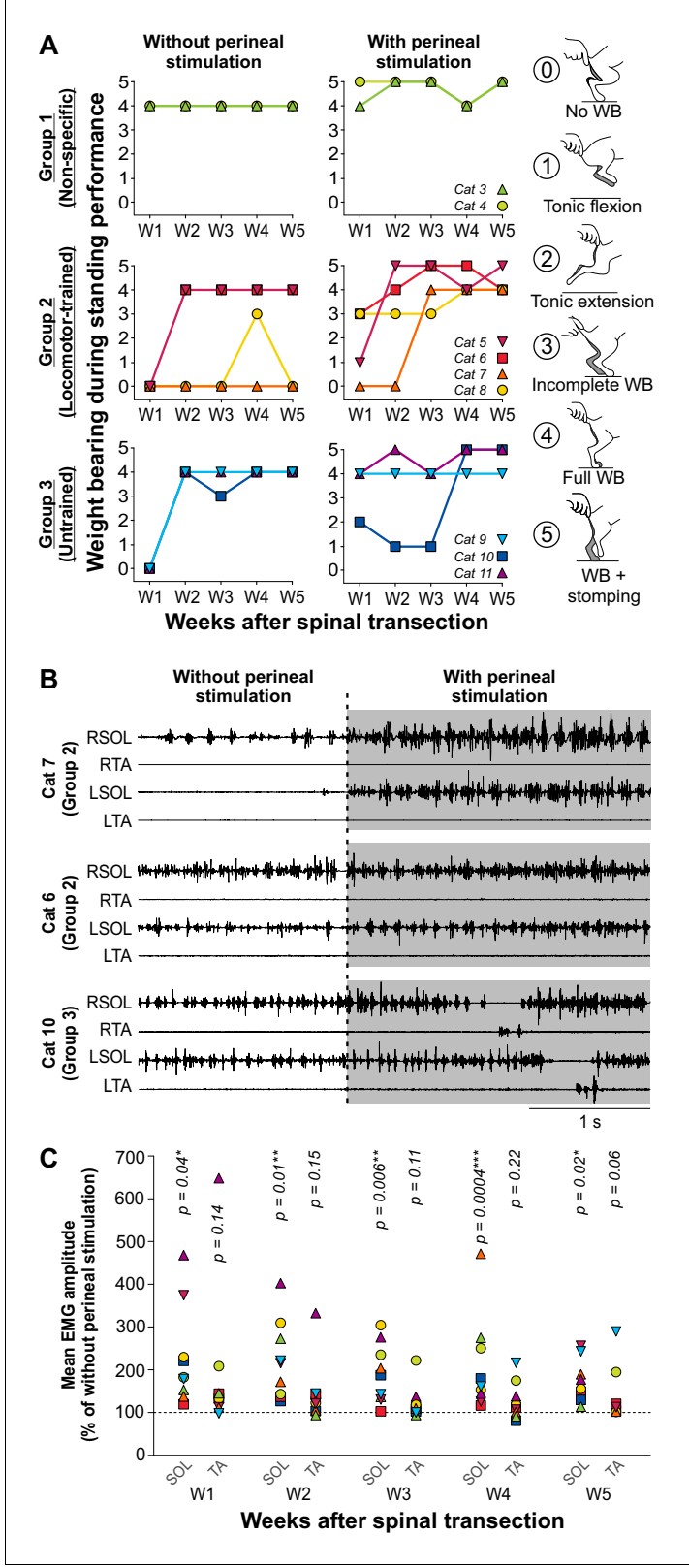

**Figure 2.** Recovery of weight bearing during standing after spinal transection. (**A**) Weight bearing during standing performance at weeks 1 to 5 (W1–W5) after spinal transection in nine individual cats without and with perineal stimulation using a 6-point scale (right panels). (**B**) Electromyography (EMG) of the right (R) and left (L) soleus (SOL) and tibialis anterior (TA) muscles without and with perineal stimulation (gray area) in three cats during standing five

*Figure 2 continued on next page*

*Figure 2 continued*

weeks after spinal transection. (C) Effect of perineal stimulation on the mean EMG amplitude of SOL and TA at weeks 1 to 5 after spinal transection of 9 individual cats obtained during 1 s of weight bearing with perineal stimulation expressed as a percentage of the amplitude obtained without perineal stimulation. P values from paired t-tests are indicated above the data points. *, p<0.05; **, p<0.01; ***, p<0.001.

perineal stimulation. Therefore, results indicate that locomotor recovery after complete SCI does not require task-specific training and largely occurs spontaneously.

To determine the effects of perineal stimulation on locomotor activity, we measured cycle and phase durations along with the burst durations and mean EMG amplitude of selected muscles without and with perineal stimulation at week six after spinal transection. *Figure 4A* shows the EMG activity of four hindlimb muscles, the hip flexor/knee extensor anterior sartorius (SRT) and one extensor muscle bilaterally (BFA, biceps femoris anterior; SOL, soleus; VL, vastus lateralis), along with the support phases in the intact state and six weeks after spinal transection without and with perineal stimulation in four cats. If cats could step without perineal stimulation but with relatively weak hindlimb muscle activity, such as Cat 3, perineal stimulation increased the activity of flexors and extensors, restoring it to levels similar to the intact state, and produced a more robust locomotor pattern (*Video 7*). On the other hand, if the locomotor pattern was robust without perineal stimulation, perineal stimulation had a slightly detrimental (Cat 6) or negligible (Cat 9) effect on hindlimb locomotion and reduced EMG activity in the hip flexor SRT (*Video 7*). In cats that displayed no locomotor activity without perineal stimulation, such as Cat 7, although perineal stimulation increased flexor and extensor activity bilaterally, the hindlimb locomotor pattern remained disorganized, with an inability to flex the hip and move the limb forward during swing (*Video 7*).

As with weight bearing during standing, the effects of perineal stimulation on hindlimb locomotion varied between animals. Thus, we pooled data across cats because there were no visible differences between experimental groups. Across cats, perineal stimulation had no significant effect on cycle, stance or swing durations or on the phasing between hindlimbs at week six after spinal transection, with individual cats showing an increase or decrease in these temporal parameters (*Figure 4B*, paired t-tests). Although perineal stimulation had no significant effect on the burst duration of triceps surae (TS) muscles, due to increases or decreases depending on the animal, it significantly increased mean EMG amplitude (*Figure 4C*, two leftmost panels; paired t-tests). On the other hand, perineal stimulation did not significantly affect the burst duration or mean EMG amplitude of the SRT muscle, again due to some cats showing either an increase or decrease (*Figure 4C*, two rightmost panels; paired t-tests). Thus, 6 weeks after spinal transection, perineal stimulation improved locomotor performance in cats with weak locomotor activity, seemingly by augmenting extensor activity, whereas it had a negligible effect on cats with an already robust locomotor pattern.

## The recovery of speed-dependent modulation does not require specific training after spinal transection

To determine if the ability to perform hindlimb locomotion at different speeds requires specific training, we tested cats at treadmill speeds of up to 1.0 m/s at week six after spinal transection. In the nine cats that we tested treadmill locomotion each week after spinal transection, the testing period lasted ~30 s and was done only at 0.4 m/s, whereas in the other three cats, they were not placed on the treadmill at all until week six after spinal transection. In Group 2, the Locomotor-trained group, cats were only trained at a treadmill speed of 0.4 m/s. However, at week six after spinal transection, all cats that recovered hindlimb locomotion (n = 10/12 cats) could also adjust their pattern up to a maximal speed of 0.8 m/s (n = 1 cat) or 1.0 m/s (n = 9 cats) with perineal stimulation in the tied-belt condition (*Table 1*). Without perineal stimulation, six cats, including at least one from each group, attained a maximal speed of 1.0 m/s in the tied-belt condition (*Table 1*). *Figure 5A* shows the EMG activity of four hindlimb muscles, the SRT and one extensor muscle (BFA or VL) bilaterally, along with the support phases in the intact state and at week six after spinal transection with perineal stimulation in three cats, one from each experimental group. As can be observed, all three cats adjusted their hindlimb locomotor pattern to an increase in treadmill speed from 0.4 m/s to 1.0 m/s (*Video 8*). The ability to modulate cycle and phase durations and the phasing between hindlimbs as a function

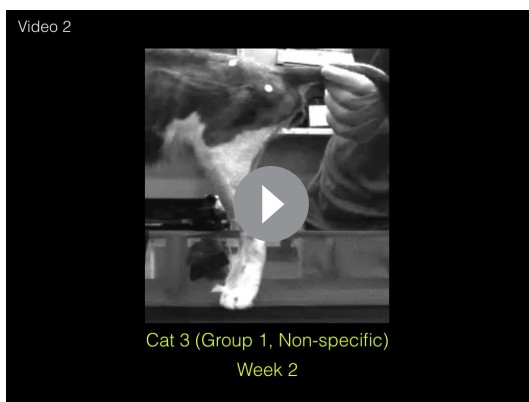

Video 2

Cat 3 (Group 1, Non-specific)
Week 2

**Video 2.** Recovery of full weight bearing standing without perineal stimulation two weeks after spinal transection. The video shows the recovery of full weight bearing standing two weeks after spinal transection in three cats, one from each experimental group.
https://elifesciences.org/articles/50134#video2

of treadmill speed was not visibly different for the three groups (*Figure 5B*). Across cats, an increase in speed significantly decreased cycle (p=$2.64 \times 10^{-6}$) and stance (p=$7.61 \times 10^{-7}$) durations, with no significant effects on swing duration (p=0.11) and the phasing between hindlimbs (p=0.54) (one-factor ANOVA).

Burst durations from the VL or BFA, which are active during most of the stance phase, significantly decreased with increasing speed across cats (p=$2.07 \times 10^{-6}$, one-factor ANOVA), with no visible differences between the three groups (*Figure 6*, top panel). On the other hand, SRT burst duration, which is active during most of the swing phase, was not significantly modulated by increasing speed across cats (p=0.97, one-factor ANOVA), with no visible differences between the three groups (*Figure 6*, 2nd panel from top). An increase in treadmill speed did not significantly increase the mean EMG amplitude of the extensor muscles VL or BFA (p=0.08, one-factor ANOVA) but we did observe a significant increase in the hip flexor SRT (p=0.05, one-factor ANOVA), due to variable changes with increasing speed between animals (*Figure 6*, bottom two panels). Thus, overall, spinal-transected cats recovered the ability to adjust their pattern to increasing speed without requiring training at different speeds.

## The ability to perform split-belt locomotion does not require task-specific training after spinal transection

To determine if a locomotor task that was not trained for recovered after spinal transection, we tested hindlimb locomotion on a split-belt treadmill, where the speeds of the two belts differed, at week six after spinal transection. Studies have

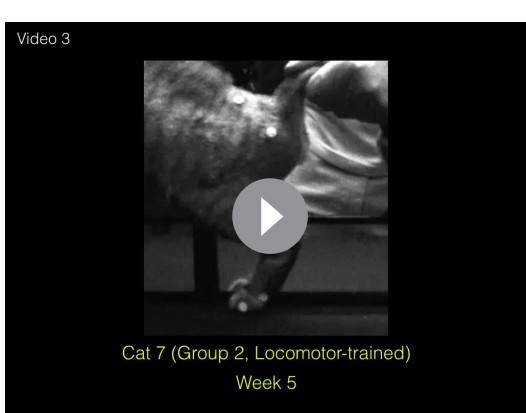

Video 3

Cat 7 (Group 2, Locomotor-trained)
Week 5

**Video 3.** Some cats did not recover the capacity to stand after spinal transection. The video shows an example of a cat (Cat 7) that did not recover standing five weeks after spinal transection without perineal stimulation.
https://elifesciences.org/articles/50134#video3

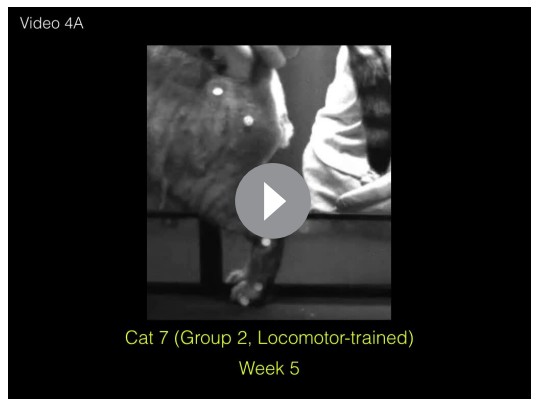

Video 4A

Cat 7 (Group 2, Locomotor-trained)
Week 5

**Video 4.** The effects of perineal stimulation on standing five weeks after spinal transection. The video shows three examples of the effects of stimulating the perineal region after spinal transection five weeks after spinal transection in three cats. (A) Perineal stimulation restored weight bearing during standing. (B) In a cat that had weight bearing during standing, perineal stimulation slightly elevated the pelvis by extending the hindlimbs. (C) In a cat that had weight bearing during standing, perineal stimulation generated stomping in place.
https://elifesciences.org/articles/50134#video4

shown that spinal-transected cats can perform locomotion on a split-belt treadmill (*Forssberg et al., 1980*; *Frigon et al., 2013*; *Frigon et al., 2017*; *Desrochers et al., 2019*). In the present study, cats received no testing or training on the split-belt treadmill before week six after spinal transection. During split-belt locomotor testing, we set the speed of the slow belt at 0.4 m/s and the speed of the fast belt at 0.5 m/s and increased it to 1.0 m/s in 0.1 m/s increments. We have shown that over this range of treadmill speeds for the slow and fast limbs, spinal-transected cats maintain a 1:1 coordination, defined as equal cycle duration for both hindlimbs (*Frigon et al., 2013*; *Frigon et al., 2017*; *Desrochers et al., 2019*). In the 10 cats that recovered hindlimb locomotion, we tested adjustments to split-belt locomotion in all of them, although some were not tested without perineal stimulation (*Table 1*). All nine cats, including four cats from Group 1, two cats from Group 2 and 3 cats from Group 3 performed split-belt locomotion with a maximal speed of 0.8 m/s (n = 1 cat) or 1.0 m/s (n = 8 cats) for the fast limb, without or with perineal stimulation (*Table 1*). Note that the hindlimb locomotor pattern in some cats adjusted better if it was the left or right hindlimb stepping on the fast belt. Perineal stimulation generally facilitated split-belt locomotion.

*Figure 7A* shows the EMG activity of four hindlimb muscles, the SRT and one extensor muscle (BFA or VL) bilaterally, along with the support phases at week 6 after spinal transection in three cats, one from each experimental group, during split-belt locomotion with the slow left hindlimb at 0.4 m/s and the right fast hindlimb stepping at 0.5 m/s, 0.7 m/s and 1.0 m/s. Cat 2 and Cat 5 stepped with perineal stimulation while Cat 9, from the Untrained Group, stepped without perineal stimulation (*Video 9*). As can be observed, all three cats adjusted their hindlimb locomotor pattern to an increase in the treadmill speed of the fast belt from 0.5 m/s to 1.0 m/s. To maintain a 1:1 coordination, the proportion of the stance phase of the slow limb and the swing phase of the fast limb increased, as shown in intact, decerebrate and spinal-transected cats as well as healthy humans (*Kulagin and Shik, 1970*; *Halbertsma, 1983*; *Dietz et al., 1994*; *Reisman et al., 2005*; *Frigon et al., 2015*; *Frigon et al., 2017*).

The ability to modulate cycle and phase durations and the phasing between hindlimbs as a function of treadmill speed was not visibly different for the three groups for the fast and slow hindlimbs (*Figure 7B*). Across cats, an increase in speed significantly decreased cycle (p=0.001), stance (p=$2.54\times10^{-13}$) and swing (p=$1.02\times10^{-4}$) durations of the fast limb, with no significant effect on the phasing between hindlimbs (p=0.45) (one-factor ANOVA). For the slow hindlimb stepping at 0.4 m/s, increasing the speed of the treadmill of the fast belt significantly increased cycle (p=0.004) and swing durations (p=0.01) with no significant effect on stance duration (p=0.72) (one-factor ANOVA). Note that we did not test all speeds in Cat 11 of Group 3. Therefore, these results indicate that spinal-transected cats recover the ability to perform split-belt locomotion, maintaining interlimb coordination, without requiring task-specific training.

## Discussion

### Locomotor recovery after spinal cord injury does not require task-specific training

Consistent with our hypothesis, we found that the recovery of hindlimb locomotion after complete SCI in the adult cat does not require task-specific training. Indeed, all cats that did not receive task-specific locomotor training, which includes 4 of 4 cats in Group 1 (Non-specific) that received manual therapy and 4 of 4 cats in Group 3 (Untrained) that received no intervention or training of any kind after spinal transection recovered hindlimb locomotion (*Figure 3*). Moreover, 3 cats out of 4 in Group 1 attained the highest performance level on our scale without and with perineal stimulation 6 weeks after spinal transection, whereas 2 and 3 cats attained the highest performance level without and with perineal stimulation, respectively, in Group 3. It should be emphasized that three cats, Cat 1 and Cat 2 from Group 1 and Cat 12 from Group 3 were not tested for standing or locomotor performance in the five weeks after spinal transection. At week 6 after spinal transection, on their very first testing day, these three cats performed hindlimb locomotion at the highest level on our scale, which includes full weight bearing and proper digitigrade placement (*Figure 3*). Therefore, we showed that hindlimb locomotion recovered with rhythmic manual stimulation of the triceps surae muscles, a non-task-specific training (Group 1, Non-specific), as well as without any intervention (Group 3, Untrained). In other words, our results indicate that task-specific sensory feedback does

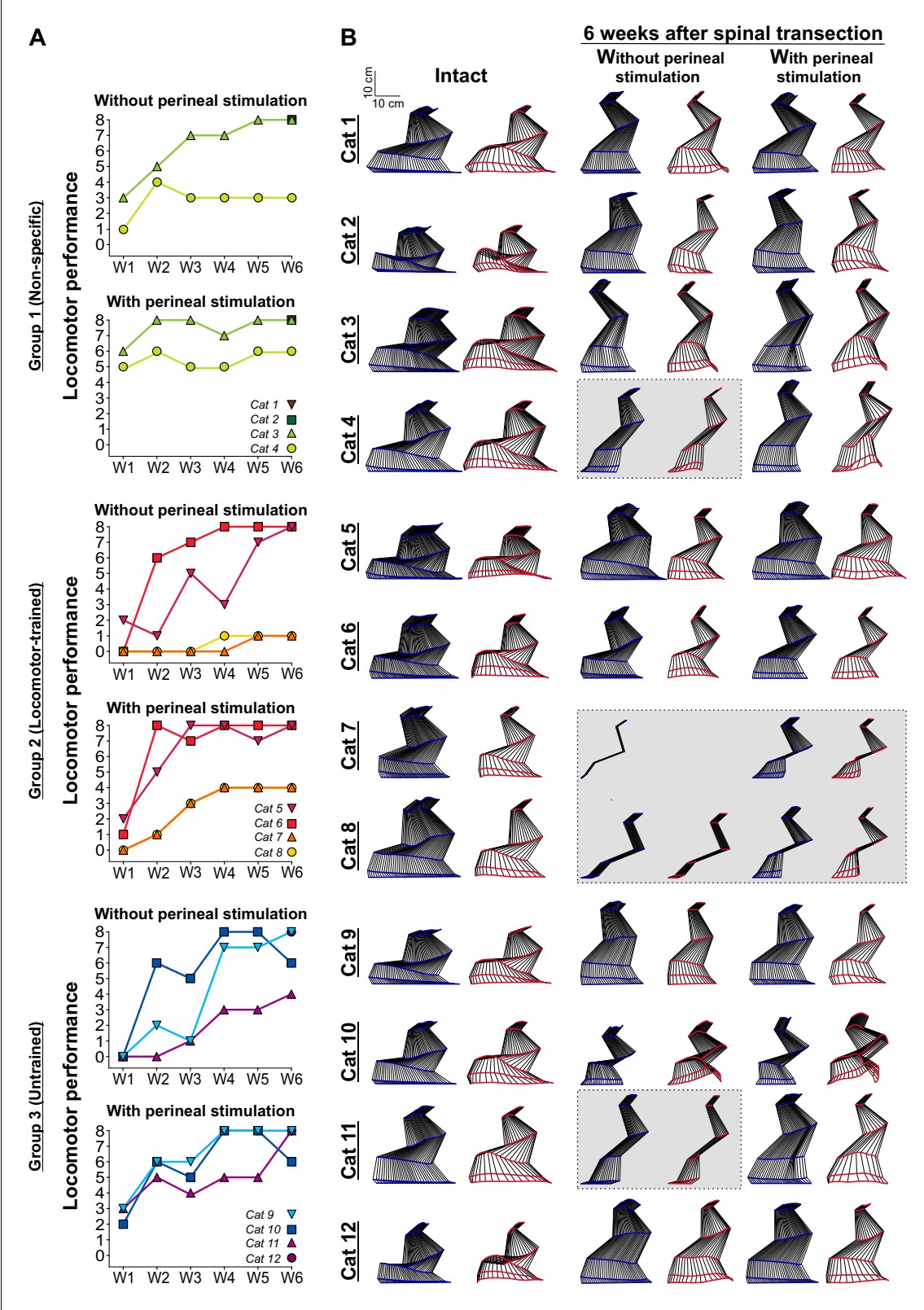

**Figure 3.** Recovery of hindlimb locomotion after spinal transection. (A) Hindlimb locomotor performance during tied-belt locomotion at 0.4 m/s at weeks 1 to 6 (W1–W6) after spinal transection in nine (W1–W5) and twelve (W6) individual cats without and with perineal stimulation using a nine-point scale. (B) A stick figure diagram of a representative cycle showing kinematics of the right hindlimb without and with perineal stimulation before (Intact) and six weeks after spinal transection during tied-belt locomotion at 0.4 m/s in the twelve cats. Gray areas indicate animals that could not step.

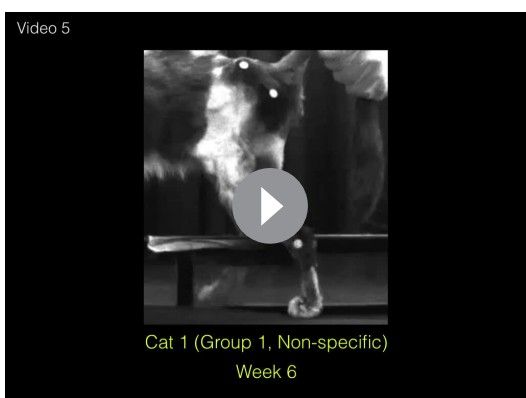

**Video 5.** Recovery of full weight bearing hindlimb locomotion six weeks after spinal transection. The video shows the recovery of full weight bearing hindlimb locomotion without perineal stimulation six weeks after spinal transection at a treadmill speed of 0.4 m/s in three cats, one from each experimental group.

https://elifesciences.org/articles/50134#video5

not drive hindlimb locomotor recovery after complete SCI. The notion that mass practice of locomotion is beneficial to locomotor recovery in preclinical models of complete SCI has become widely accepted and current rehabilitation approaches in people with SCI include several weeks or months of intensive BWSTT (*Angeli et al., 2018*; *Gill et al., 2018*; *Wagner et al., 2018*). However, before these recent studies in people with SCI, a few studies had shown that spinal-transected animals in control groups, receiving no treatment or non-task-specific training, such as stand training, recovered some level of hindlimb locomotor performance (*Lovely et al., 1986*; *Hodgson et al., 1994*; *De Leon et al., 1998a*; *De Leon et al., 1998b*; *De Leon et al., 1999b*). These studies emphasized that treadmill-trained animals had a tendency to recover better than controls, thus obscuring the message that hindlimb locomotion recovered without task-specific locomotor training. In the present study, we clearly show that spinal-transected cats receiving manual therapy

**Table 1.** Locomotor performance of individual cats six weeks post-transection without and with perineal stimulation.
Individual cats are listed on the left along with the group (G) they belonged to and whether they were female (F) or male (M). The table shows several locomotor parameters for individual cats of the three groups, including the maximal speed during tied-belt locomotion, tested up to 1.0 m/s, and the maximal speed of the fast limb during split-belt locomotion. For split-belt locomotion, the slow limb stepped at 0.4 m/s while the left (L) and right (R) hindlimbs were tested up to a maximal speed of 1.0 m/s. NT, Not tested; Y, proper digitigrade placement of the paw at contact; I, inconsistent paw placement. A dash mark indicates an inability to perform hindlimb locomotion.

| | Without perineal stimulation | | | | With perineal stimulation | | | |
|---|---|---|---|---|---|---|---|---|
| Cat | Maximal speed (m/s) Tied-belt | Maximal speed (m/s) Split-belt | # of consecutive steps | Proper Paw placement | Maximal speed (m/s) Tied-belt | Maximal speed (m/s) Split-belt | # of consecutive steps | Proper Paw placement |
| 1-G1 (F) | 1.0 | L: 1.0; R: 1.0 | >10 | Y | 1.0 | L: 1.0; R: 1.0 | >10 | Y |
| 2-G1 (F) | 0.4-NT | NT | >10 | Y | 1.0 | L: 1.0; R: 1.0 | >10 | Y |
| 3-G1 (M) | 0.0 | / | / | / | 1.0 | L: 1.0; R: 1.0 | <10 | I |
| 4-G1 (M) | 0.4-NT | NT | >10 | Y | 1.0 | L: 1.0; R: 1.0 | >10 | Y |
| 5-G2 (F) | 1.0 | L:0.7; R: 1.0 | >10 | Y | 1.0 | L: 1.0; R: 1.0 | >10 | Y |
| 6-G2 (F) | 1.0 | L: 1.0; R: 1.0 | >10 | Y | 0.8 | NT | >10 | Y |
| 7-G2 (F) | 0.0 | / | / | / | 0.0 | / | / | / |
| 8-G2 (M) | 0.0 | / | / | / | 0.0 | / | / | / |
| 9-G3 (F) | 1.0 | L: 0.7; R: 1.0 | >10 | Y | 1.0 | L: 1.0; R: 1.0 | >10 | Y |
| 10-G3 (F) | 1.0 | L: 0.8; R: 0.8 | >10 | Y | 1.0 | L: 0.8; R: 0.8 | >10 | Y |
| 11-G3 (M) | 0.0 | / | / | / | 1.0 | L: 1.0; R: 1.0 | >10 | I |
| 12-G3 (M) | 0.4-NT | NT | <10 | I | 1.0 | L: 0.9; R: 0.9 | >10 | I |

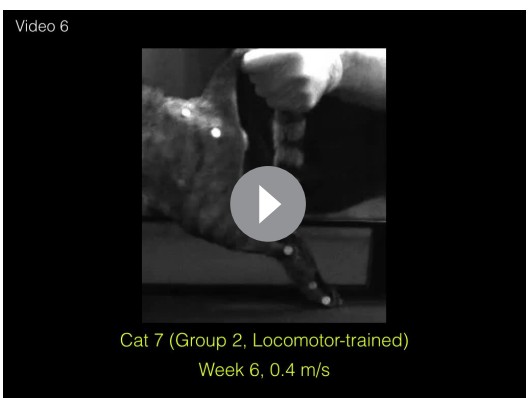

Video 6 Cat 7 (Group 2, Locomotor-trained) Week 6, 0.4 m/s

**Video 6.** Some cats did not recover hindlimb locomotion after spinal transection. The video shows an example of a cat (Cat 7) from the Locomotor-trained group that did not recover hindlimb locomotion without perineal stimulation six weeks after spinal transection. The treadmill speed was set at 0.4 m/s. https://elifesciences.org/articles/50134#video6

or no training at all recovered a high level of performance six weeks after complete SCI, which included locomotion from 0.1 m/s to 1.0 m/s and also split-belt locomotion with the fast hindlimb stepping at more than twice the speed of the slow hindlimb. Moreover, three animals, two from the Non-specific group and one from the Untrained group, were not even tested on the treadmill until week 6 after spinal transection, and these animals recovered hindlimb locomotion to our highest performance level. Our results are consistent with a study in spinal-transected rats that found no significant differences between trained and untrained groups (*Alluin et al., 2015*). Together, these results indicate that the spinal locomotor CPG located at lumbar levels and its interactions with sensory feedback from the limbs are largely intact after complete SCI and do not require task-specific neuroplastic mechanisms to regain their functionality. Additionally, the ability to modulate speed and coordinate left-right activity during split-belt locomotion is conserved after complete SCI, without requiring specific training to recover.

The observation that hindlimb locomotion recovered without training in our Untrained group suggests that it occurred spontaneously. However, we observed that most cats displayed sporadic episodes of rhythmic activity of their hindlimbs when sitting in their cages while they were supporting their body weight with their forelimbs. This activity consisted of v-shaped alternating forward and backward hindlimb movements. In this sitting position, the perineal skin is stimulated and likely activates the spinal locomotor CPG. Some cats also displayed vigorous rhythmic muscle spasms, consistent with the presence of spasticity (*Roy and Edgerton, 2012*). Thus, it is possible that these rhythmic activities participated in the recovery of hindlimb locomotion by providing excitatory inputs to spinal sensorimotor circuits. However, these types of activities are non-task-specific and do not change our main finding that task-specific training is not required for locomotor recovery. It is also possible that the emergence of these rhythmic activities were a consequence, and not the cause, of a return of excitability within spinal sensorimotor locomotor circuits.

We also showed that weight bearing during standing is a function that recovered without task-specific training, and in all cats that we tested weekly after spinal transection, it recovered before hindlimb locomotion. It was shown that training spinal-transected cats to stand interfered with the recovery of hindlimb locomotion, whereas treadmill training did not promote standing recovery (*Hodgson et al., 1994*). Although we did not compare stand training to step training, we show that the ability to stand and step recover in parallel, with weight bearing during standing recovering more rapidly than hindlimb locomotion.

## Locomotor recovery after spinal cord injury requires sufficient excitability within the spinal sensorimotor circuits

To increase spinal neuronal excitability, we manually stimulated the perineal skin during standing and during treadmill locomotion, which is known to facilitate hindlimb locomotion in spinal-transected mammals through an undefined mechanism (*Barbeau and Rossignol, 1987*; *Bélanger et al., 1996*; *Leblond et al., 2003*; *Alluin et al., 2015*). With perineal stimulation, full hindlimb weight bearing during standing was present as early as one week after spinal transection and the highest level of hindlimb locomotor performance was attained as early as two weeks after transection in some cats (*Figures 2A* and *3A*). During standing and locomotion, perineal stimulation increased the activity of extensors, with no significant increase in flexors (*Figures 2* and *4*). Over time, standing and hindlimb locomotion recovered without the need for the facilitation of spinal neuronal excitability provided by perineal stimulation. In other words, the re-expression of hindlimb locomotion after complete SCI required the return of neuronal excitability in largely intact spinal sensorimotor circuits,

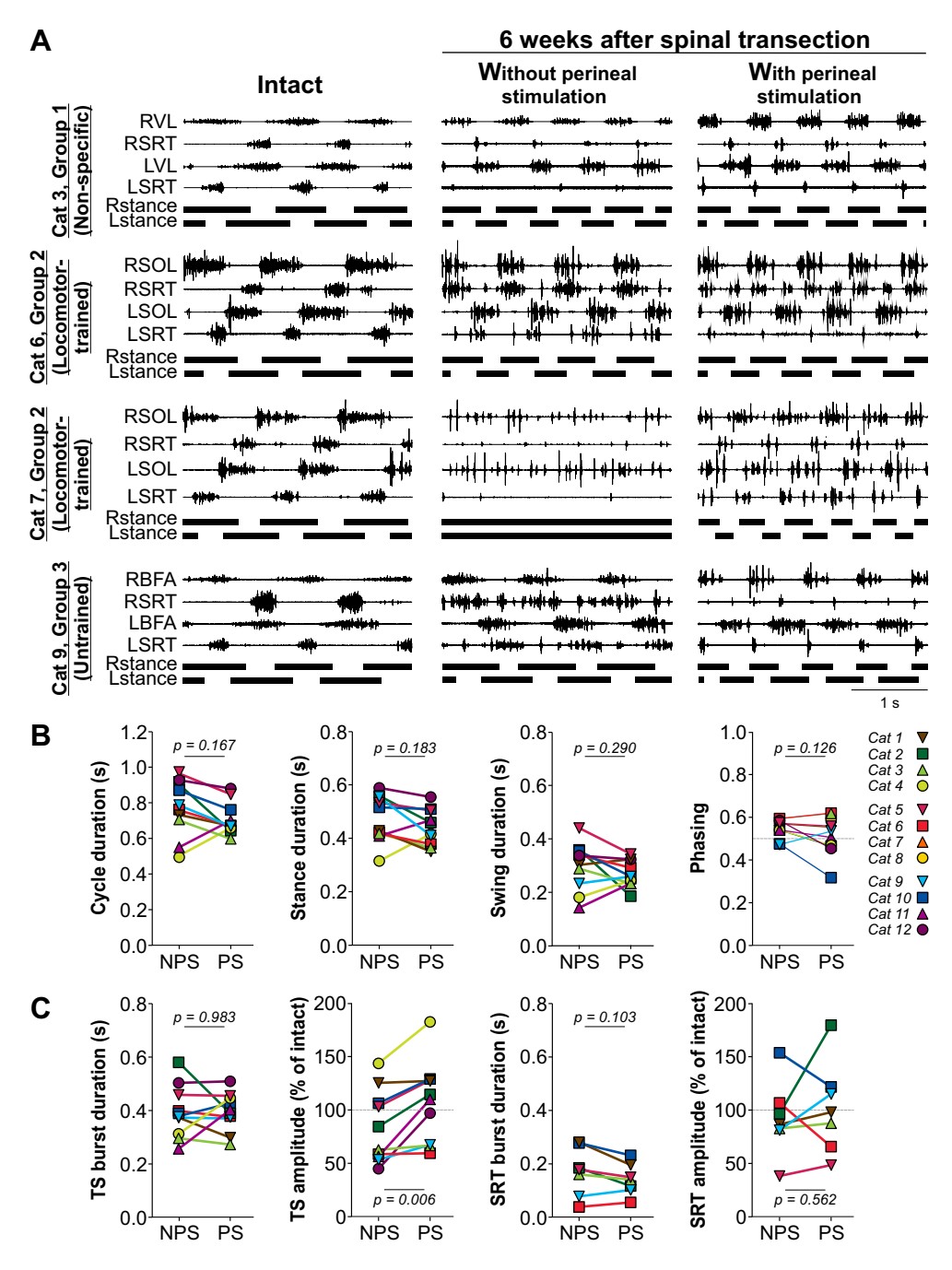

**Figure 4.** Hindlimb muscle activity during locomotion before and after spinal transection. (**A**) Hindlimb locomotor pattern before (Intact) and six weeks after transection in four cats from the three groups, including two from Group 2, during tied-belt locomotion at 0.4 m/s. The effects of perineal stimulation is shown after spinal transection. Each panel shows the EMG from four hindlimb muscles from the right (R) and left (L) hindlimbs: SOL, soleus; BFA, biceps femoris anterior; SRT, anterior sartorius. (**B**) Cycle, stance and swing durations and the phasing between hindlimbs with no (NPS) or with (PS) perineal stimulation at 6 weeks after spinal transection. (**C**) Effect of perineal stimulation on the burst durations and mean EMG amplitudes of the triceps surae (TS, soleus n = 8 or lateral gastrocnemius n = 2) or SRT (n = 7) muscles at 6 weeks after spinal transection. P values above panels in B and C from paired t-tests comparing values obtained without and with perineal stimulation.

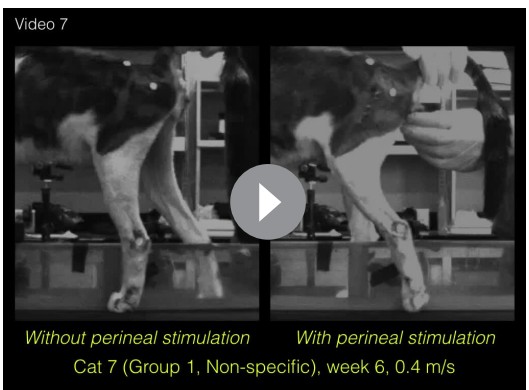

Video 7. The effects of perineal stimulation on hindlimb locomotion six weeks after spinal transection. The video shows the effects of perineal stimulation on four cats, one from Groups 1 (Non-specific) and 3 (Untrained) and two from Group 2 (Locomotor-trained), on hindlimb locomotion at a treadmill speed of 0.4 m/s six weeks after spinal transection. Left and right panels respectively show hindlimb locomotion without and with perineal stimulation.

https://elifesciences.org/articles/50134#video7

which is normally facilitated by monoaminergic inputs from the brainstem. Our results are consistent with several pharmacological studies that have shown that increasing spinal neuronal excitability by injecting noradrenergic agonists in spinal-transected cats (*Forssberg and Grillner, 1973*; *Barbeau et al., 1987*; *Chau et al., 1998a*; *Chau et al., 1998b*) or serotonergic agonists in spinal-transected rodents (*Feraboli-Lohnherr et al., 1999*; *Kim et al., 2001*; *Antri et al., 2002*; *Antri et al., 2005*; *Sławińska et al., 2012*) facilitated hindlimb locomotion. In a similar vein, reducing inhibition by blocking GABAergic or glycinergic transmission within the spinal cord of spinal-transected animals facilitated hindlimb locomotion (*Robinson and Goldberger, 1986*; *De Leon et al., 1999b*). Taken together, these results indicate that spinal locomotor sensorimotor circuits are functional after complete SCI but lack sufficient excitability. Providing excitability restores hindlimb locomotion without the need for weeks of intensive task-specific locomotor training.

Spinal transection is undoubtedly the most reproducible type of SCI in pre-clinical models. Despite this high reproducibility, the recovery of standing and hindlimb locomotion in spinal-transected cats varied between animals (*Figures 2* and *3*). The inter-animal variability in recovery can be due to several factors, including inherently different potentials for recovery, the amount of episodes of rhythmic non-locomotor activities and the general health of the animal. Another factor to consider is whether the sex of the animals impacted locomotor recovery after SCI. We think that it is unlikely that sex played a role in locomotor recovery because the Non-specific and the Untrained groups showed similar levels of recovery and included 2 males and 2 females in each group. The sex of individual cats is indicated in *Table 1*. Although, the Locomotor-trained group included 3 females and 1 male, the two cats that did not recover hindlimb locomotion included a male and a female cat. We do not know why these two animals did not recover hindlimb locomotion after spinal transection. Although we can only speculate, their inability to perform hip placement for rostral positioning of the paw and their lack of weight bearing suggests failure in spinal circuits that coordinate motor pools and/or integrate sensory feedback from the limbs.

## Concluding remarks

In the present study, we asked the following question: does sensory feedback need to be specific to the task to facilitate the recovery of hindlimb locomotion after complete SCI? The answer is no. Sensory feedback is undoubtedly required to initiate and control hindlimb locomotion in spinal-transected cats, allowing it to adjust to different treadmill speeds during tied-belt and split-belt locomotion. However, during the recovery period, task-specific sensory feedback is not required to drive the recovery of hindlimb locomotion, speed modulation or the ability to adjust to split-belt locomotion.

The neurophysiological basis for locomotor training in pre-clinical animal models and in humans is scientifically sound. It assumes that the spinal locomotor CPG located at lumbar levels caudal to the low thoracic SCI is largely intact and that it can interact with sensory feedback from the limbs. It was suggested that these interactions through mass practice of walking supposedly teach the spinal locomotor CPG to function with diminished inputs from the brain. The evidence for a spinal locomotor CPG in humans has been extensively presented and discussed (*Duysens and Van de Crommert, 1998*; *Dietz, 2003*; *Yang et al., 2004*; *Minassian et al., 2017*) and studies have shown that the injured spinal cord strongly responds to sensory feedback from the limbs (*Hodgson et al., 1994*;

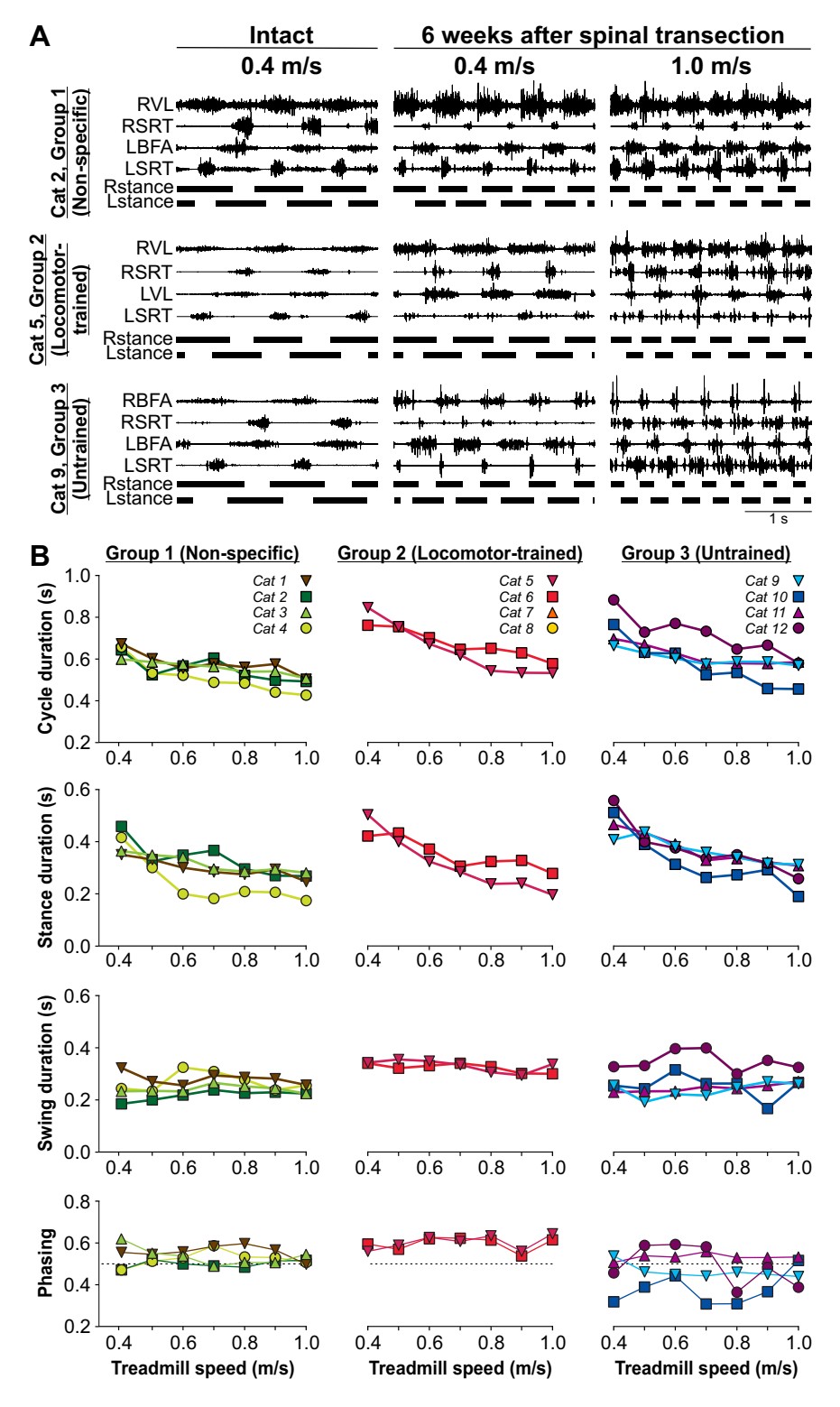

**Figure 5.** Speed modulation during tied-belt locomotion before and six weeks after spinal transection. (A) Hindlimb locomotor pattern before (Intact) and six weeks after spinal transection in three cats, one from each group, during tied-belt locomotion at 0.4 m/s and at 1.0 m/s in the spinal state. In the examples shown, cats stepped with perineal stimulation. Each panel shows the EMG from four hindlimb muscles from the right (R) and left (L) hindlimbs: BFA, biceps femoris anterior; SRT, anterior sartorius; VL, vastus lateralis. (B) Cycle, stance and

*Figure 5 continued on next page*

*Figure 5 continued*

swing durations and the phasing between hindlimbs. All cats stepped with perineal stimulation except for Cat 6. Each data point is the mean of 10–15 cycles.

---

*Harkema et al., 1997*; *Dietz et al., 2002*; *Dy et al., 2010*). We know from animal and human studies that somatosensory feedback is essential for the expression of locomotion after SCI (*Bouyer and Rossignol, 2003*; *Takeoka et al., 2014*; *Bui et al., 2016*; *Formento et al., 2018*). This is not surprising because humans with an intact central nervous system cannot walk following the loss of touch and proprioceptive feedback and rare cases of recovery require years of intense therapy, as all movements need to be consciously planned and executed (*Lajoie et al., 1996*). Spinal cord injury disrupts many proprioceptive and cutaneous pathways ascending to the brain that are essential for the control of posture and locomotion. In other words, people with SCI must cope not only with less voluntary control over their muscles but also with absent or less detailed information about their movements.

The results from our Locomotor-trained group, in which two cats did not recover the capacity for standing or hindlimb locomotion after complete SCI, suggest that locomotor training might even be detrimental. However, we do not think this is the case. Hindlimb locomotor performance in spinal-transected cats runs along a continuum with low-level to high-level performers, regardless of the approach used in the recovery period (*Figure 3*). We think it a coincidence that the two cats that did not recover hindlimb locomotion fell into the Locomotor-trained group, as the other two cats in this group recovered standing and locomotion to the highest level without perineal stimulation. Locomotor training in humans is a form of exercise that undoubtedly provides benefits, such as promoting cardiovascular and musculoskeletal health (*Yang et al., 2004*; *Maher et al., 2017*; *van der Scheer et al., 2017*), as well as providing social interactions. However, whether locomotor training is an optimal form of exercise for inducing these benefits in people with SCI requires further investigation, as locomotor training is costly and demanding for participants, both physically and mentally.

To conclude, the results from the present study and other studies discussed indicate that the recovery of hindlimb locomotion simply requires an adequate return of excitability with spinal neuronal circuits caudal to the SCI. The important distinction to make for locomotor recovery after SCI is whether animals, including humans, learn or re-express locomotion. Motor learning can broadly be defined as an experience-dependent improvement in performance (*Krakauer et al., 2019*). However, this definition generally applies to the acquisition of new motor skills. On the other hand, the ability to perform locomotion or locomotor-like movements is present at birth, as shown in several neonatal animals, including humans (*Clarac et al., 2004*). Thus, we should consider the recovery of hindlimb locomotion in pre-clinical models a re-expression of an impaired innate function, accomplished by restoring sufficient excitability within spinal sensorimotor circuits, as opposed to learning a skill. In humans with severe SCI, however, it is entirely possible that they learn to consciously plan and execute every step, as seen in people with no touch or proprioceptive feedback. This is probably why the walking pattern in humans with severe SCI loses its automaticity (*Angeli et al., 2018*; *Gill et al., 2018*; *Wagner et al., 2018*).

## Materials and methods

### Animals and ethical information

All procedures were approved by the Animal Care Committee of the Université de

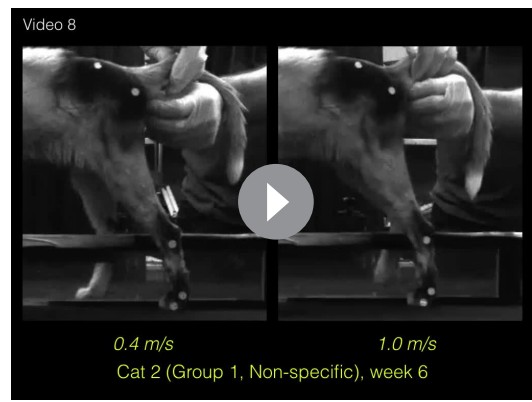

**Video 8.** Speed modulation during locomotion recovers without specific training six weeks after spinal transection. The video shows hindlimb locomotion at treadmill speeds of 0.4 m/s (left panel) and 1.0 m/s (right panel) in three cats, one from each experimental group, six weeks after spinal transection with perineal stimulation.
https://elifesciences.org/articles/50134#video8

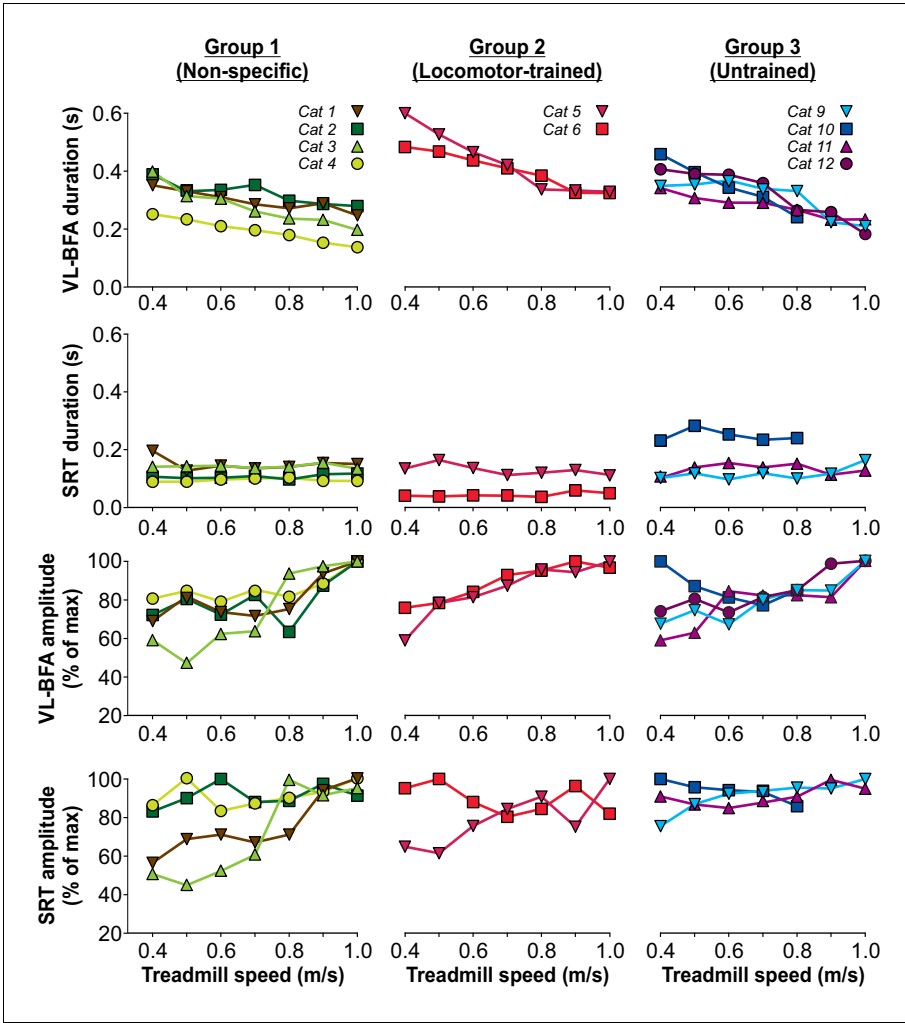

**Figure 6.** Modulation of muscle activity with increasing treadmill speed during tied-belt locomotion six weeks after spinal transection. From the top, the first panel shows burst durations of the vastus lateralis (VL, n = 4) or biceps femoris anterior (BFA, n = 6) muscles while the second panel shows burst durations of the anterior sartorius (SRT, n = 9) muscle for individual cats, separated by group, as a function of treadmill speed. The third and fourth panels show the mean EMG amplitudes of the VL-BFA and SRT, respectively, for individual cats, separated by group, as a function of treadmill speed. All cats stepped with perineal stimulation except for Cat 6. Each data point is the mean of 10–15 cycles.

Sherbrooke and were in accordance with policies and directives of the Canadian Council on Animal Care (Protocol 442–18). Twelve adult cats (>1 year of age at time of experimentation), 5 males and 7 females, weighing between 3.6 kg and 4.7 kg were used in the present study. Our study followed the ARRIVE guidelines for animal studies (*Kilkenny et al., 2010*). To reduce the number of animals used in research, some of the cats used in the present study were used in another study to answer a different scientific question (*Desrochers et al., 2019*).

## Surgical procedures

Implantation and spinal transection surgeries were performed on separate days under aseptic conditions with sterilized instruments in an operating room. Prior to surgery, butorphanol (0.4 mg/kg), acepromazine (0.1 mg/kg), and glycopyrrolate (0.01 mg/kg) were injected intramuscularly for sedation. Ketamine and diazepam (0.05 ml/kg) were then injected intramuscularly for induction. Cats were anesthetized with isoflurane (1.5–3%) delivered in O2, first with a mask and then with an endotracheal tube. During surgery, we maintained anesthesia by adjusting isoflurane concentration as

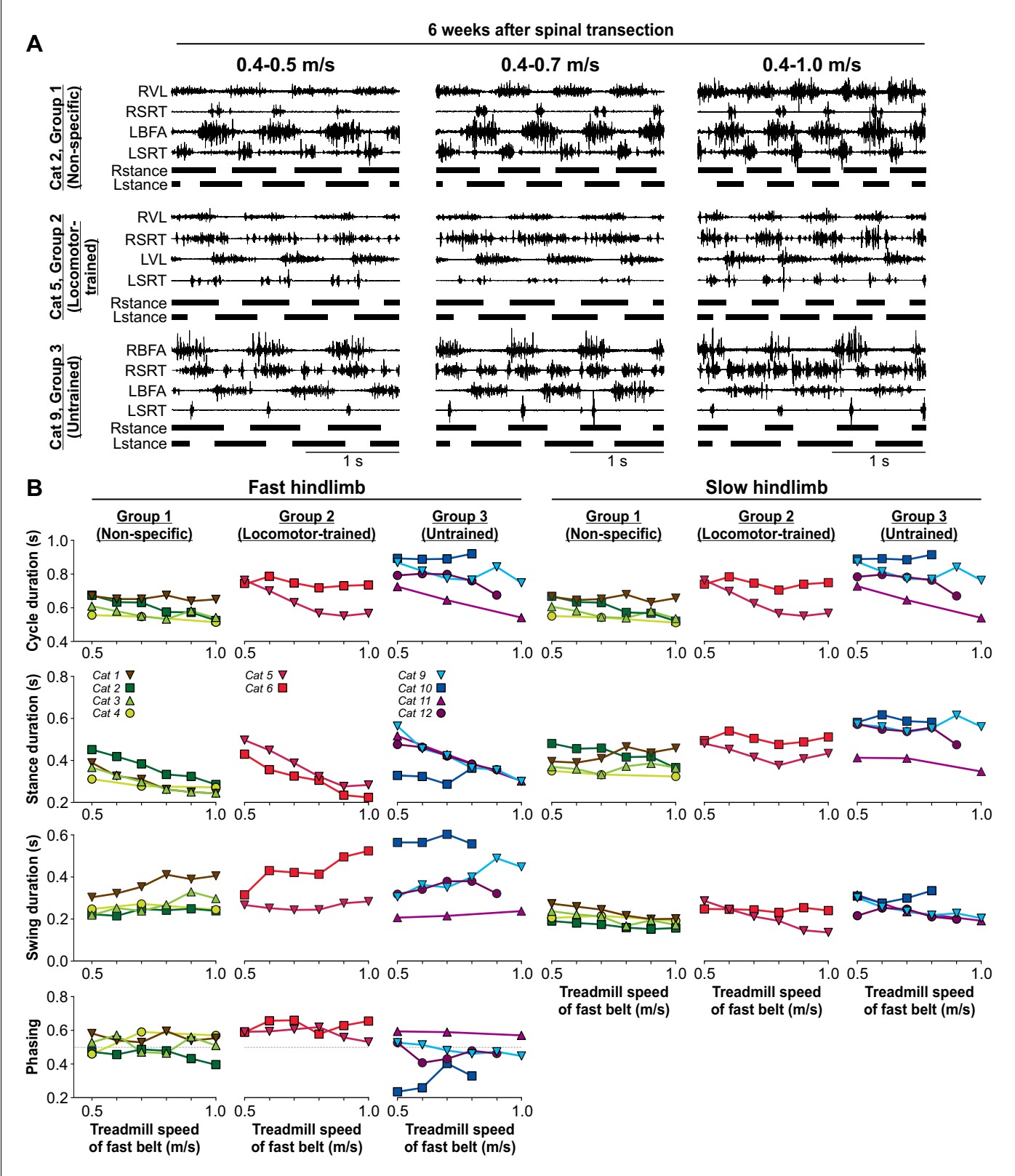

**Figure 7.** Split-belt locomotion six weeks after spinal transection. (**A**) Hindlimb locomotor pattern six weeks after spinal transection in three cats, one from each group during split-belt locomotion with the slow (left) hindlimb stepping at 0.4 m/s and the fast (right) hindlimb stepping at 0.5 m/s, 0.7 m/s and at 1.0 m/s. In the examples shown, Cat 2 and Cat 5 stepped with perineal stimulation while Cat 9 stepped without. Each panel shows the EMG

*Figure 7 continued on next page*

*Figure 7 continued*

from 4 hindlimb muscles from the right (R) and left (L) hindlimbs: BFA, biceps femoris anterior; SRT, anterior sartorius; VL, vastus lateralis. (B) Cycle, stance and swing durations and the phasing between hindlimbs for the fast (left panels) and slow (right panels) limbs. All cats stepped with perineal stimulation except for Cats 6, 9 and 10. Each data point is the mean of 10–15 cycles. Note, that some intermediate speeds were not tested in Cat 11.

needed (1.5–3%) by monitoring cardiac and respiratory rate. A catheter was also placed in a cephalic vein to continuously supply Ringers lactate solution (3 ml/kg/h) for cardiovascular support. Body temperature was monitored with a rectal thermometer and maintained within physiological range (37 ± 0.5℃) using a water-filled heating pad placed under the animal and an infrared lamp placed ~50 cm over the animal. The animal's skin was carefully shaved using electric clippers and cleaned with chlorhexidine soap. The depth of anesthesia was confirmed by applying pressure to a paw (to detect limb withdrawal) and by assessing the size and reactivity of pupils. At the end of surgery, we injected an antibiotic (Convenia, 0.1 ml/kg) subcutaneously and we taped a transdermal fentanyl patch (25 mcg/h) to the back of the animal 2–3 cm rostral to the base of the tail for prolonged analgesia (4–5 day period). We also injected buprenorphine (0.01 mg/kg), a fast-acting analgesic, subcutaneously at the end of the surgery and ~7 hr later. After surgery, we placed the cats in an incubator until they regained consciousness.

### Electrode implantation

To record muscle activity (EMG, electromyography), we directed Teflon-insulated multistrain fine wires (AS633; Cooner Wire) subcutaneously from two head-mounted 34-pin connectors (Omnetics Connector). Bipolar recording electrodes were sewn into the belly of selected hindlimb muscles, with 1–2 mm of insulation removed from each wire. The head connector was secured to the skull using dental acrylic. We verified electrode placement during surgery by electrically stimulating each muscle through the appropriate head connector channel.

### Spinal transection

The skin was incised over the last thoracic vertebrae and after carefully setting aside muscle and connective tissue, a small dorsal laminectomy was made. The dura was removed and xylocaine (Lidocaine hydrochloride, 2%) was applied topically and injected within the spinal cord. We then completely transected the spinal cord with surgical scissors between the 12th and 13th thoracic vertebrae. The gap (0.5–1.0 cm) between the two cut ends of the spinal cord was then cleaned and any residual bleeding was stopped. We verified that no spinal cord tissue remained connecting rostral and caudal ends. A hemostatic agent (Spongostan) was placed within the gap, and muscles and skin were sewn back to close the opening in anatomic layers. After spinal transection, we manually expressed the cat's bladder and large intestine one to two times daily, or as needed.

### Histology

At the conclusion of the experiments, cats were deeply anesthetized, as described above, and received a lethal dose of sodium pentobarbital (120 mg/kg) through the cephalic vein. We dissected a 2 cm length of spinal cord centered around the injury and placed it in 25 ml of 4% paraformaldehyde solution (PFA in 0.1 m PBS, 4° C). After five days, the spinal cord was cryoprotected in PBS with 30% sucrose for 72 hr at 4℃.

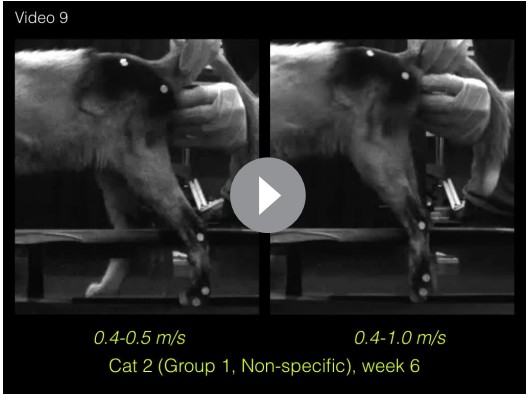

**Video 9.** The capacity for split-belt locomotion recovers without specific training six weeks after spinal transection. The video shows the hindlimb locomotor pattern on a split-belt treadmill in three cats, one from each experimental group, six weeks after spinal transection with (Cat 2 and Cat 5) or without (Cat 9) perineal stimulation. For each cat, the slow belt operated at a speed of 0.4 m/s while the fast belt was set at 0.5 m/s (left panel) and 1.0 m/s (right panel).
https://elifesciences.org/articles/50134#video9

We then cut the spinal cord in 50 μm coronal sections using a cryostat (Leica CM1860, Leica Biosystems Inc, Concord, ON, Canada). Sections were mounted on slides and stained with 1% Cresyl violet. All slides were scanned by Nanozoomer (Hamamatsu Corporation, NJ) and we performed qualitative and quantitative evaluations of the lesioned area.

## Experimental groups and recovery period

To investigate if locomotor recovery after complete SCI requires task-specific training or occurs spontaneously, we divided cats into three groups of four animals. In Group 1 (Non-specific), to evaluate non-task-specific training, cats (2 females, 2 males) received manual therapy five times a week for 5 weeks, 20 min per day, starting one week after spinal transection. Cats were placed on a padded support and a cushioned restraint stabilized the pelvis. One hindpaw was fixed to a robotic arm (Fanuc Robotics) and the other hindpaw to a rigid support (*Figure 1B*). The limb positions/joint angles were the same for both hindlimbs throughout the treatment period. For manual therapy, an experimenter applied rhythmic unilateral pressure, consisting of distal to proximal lengthwise strokes at 0.33 Hz to the triceps surae muscles. This manual therapy is akin to massage therapy applied in the clinic. Each leg was manually stimulated for 10 min, 5 times a week. In Group 2 (Locomotor-trained), to evaluate task-specific training, cats (3 females, 1 male) received locomotor training on a treadmill five times a week for 5 weeks, 20 min per day, starting one week after spinal transection. For locomotor training, the cat's hindlimbs were placed on a motorized treadmill with the forelimbs on a stationary platform. The treadmill was composed of two independently controlled running surfaces 120 cm long and 30 cm wide (Bertec, Columbus, OH). A Plexiglas separator was placed between the hindlimbs to prevent them from impeding each other. During training, two experimenters manually moved the hindlimbs at a belt speed of 0.4 m/s to simulate locomotion with appropriate joint kinematics and paw contacts. In Group 3 (Untrained), to determine if hindlimb locomotion recovers spontaneously, cats (2 females, 2 males) did not receive any treatment for the 5 weeks after spinal transection.

## Data collection and analysis

We evaluated the recovery of weight bearing during standing and hindlimb locomotion each week after spinal transection in all but three cats (two from Group 1 and one from Group 3). Chronologically, these were the first three cats experimented upon and this was done to avoid any type of training effect. However, because we observed substantial recovery in these animals, we decided to perform weekly testing for the subsequent 9 animals. To test the recovery of weight bearing during standing in the first 5 weeks after spinal transection, we collected EMG data while the hindlimbs were positioned to bear weight during standing without and with perineal stimulation for ~15 s in each condition. Perineal stimulation is known to facilitate weight bearing and hindlimb locomotion in spinal-transected animals through a non-specific increase in spinal excitability (*Barbeau and Rossignol, 1987*; *Bélanger et al., 1996*; *Leblond et al., 2003*; *Alluin et al., 2015*).

To assess locomotor recovery in the first 5 weeks after spinal transection, we started the treadmill belt at 0.4 m/s and collected EMG and kinematic data without and with perineal stimulation for ~30 s in each condition. Cats performed bipedal hindlimb locomotion with the forelimbs on a stationary platform. At week six after spinal transection, we performed a more thorough assessment during tied-belt and split-belt locomotion. In the tied-belt condition, where the two belts operated at the same speed, cats performed locomotion at speeds ranging from 0.1 to 1.0 m/s in 0.1 m/s increments. During split-belt locomotion, where the two belts operate at different speeds, the slow limb stepped at 0.4 m/s and the fast limb stepped from 0.5 to 1.0 m/s in 0.1 m/s increments. Both the left and right limbs were tested as the slow and fast limbs during split-belt locomotion. Perineal stimulation was only used for testing and not during locomotor training in Group 2. During testing, an experimenter gently held the tail to provide equilibrium. We tested weight bearing during standing and hindlimb locomotion once a week after spinal transection to avoid a training effect. To quantify the functional recovery of standing and hindlimb locomotion, we used six and nine point scales, respectively, as follows:

### Weight bearing scale during standing

1. No weight bearing

2. Tonic hip flexion
3. Tonic hip extension
4. Incomplete weight bearing
5. Full weight bearing
6. Full weight bearing plus stomping

## Locomotor scale

1. No hindlimb movements
2. Barely perceptible movements of hindlimb joints (hip, knee, ankle)
3. Tonic hyperflexion or hyperextension
4. Brisk movements at one or more hindlimb joints (hip, knee, ankle) in one or both limbs but no coordination
5. Left-right alternation but no weight bearing
6. Left-right alternation, weight bearing but improper paw placement
7. Left-right alternation, weight bearing, proper paw placement but some deficits, such as caudal paw contact and hyperflexion
8. Left-right alternation, weight bearing, proper paw placement but some deficits, such as a limping gait
9. Left-right alternation, weight bearing, proper paw placement, no visible deficits

We performed kinematic recordings as described previously (*Harnie et al., 2018*). We placed reflective markers on the skin over the iliac crest, greater trochanter, lateral malleolus, metatarsophalangeal joint and at the tip of the toes. Two cameras (Basler AcA640-100 gm) obtained videos of the left and right sides at 60 frames/s with a spatial resolution of 640 by 480 pixels. A custom-made program (Labview) acquired the images and synchronized acquisition with EMG data. By visual inspection, we determined limb contact as the first frame where the paw made visible contact with the treadmill surface, and limb liftoff as the most caudal displacement of the toe, for both hindlimbs. We used the reflective markers to perform a frame-by-frame reconstruction of hindlimb movements in a stick figure format achieved by connecting each joint sequentially. We measured cycle duration from two successive contacts of the same limb, stance duration from limb contact to liftoff and swing duration as cycle duration minus stance duration. We measured the phasing between hindlimbs as the interval of time between contact of the right and left hindlimbs normalized to cycle duration of the right hindlimb.

Electromyography was pre-amplified (x10, custom-made system), bandpass filtered (30–1000 Hz) and amplified (100-5000x) using a 16-channel amplifier (model 3500; AM Systems, Sequim, WA). As we implanted more than 16 muscles per cat, we obtained data in each locomotor condition twice, one for each connector. EMG data were digitized (2000 Hz) with a National Instruments card (NI 6032E), acquired with custom-made acquisition software and stored on computer. Although several muscles were implanted, we only describe the activity of the soleus (SOL, ankle extensor) and tibialis anterior (TA, ankle flexor) for standing experiments. For locomotion, we describe the activity of the following muscles, which were available in most cats: soleus (SOL, ankle extensor), biceps femoris anterior (BFA, hip extensor), vastus lateralis (VL, knee extensor) and the anterior sartorius (SRT, hip flexor/knee extensor). For standing, we measured EMG activity over a period of 1 s, while during locomotion we measured EMG burst activity from onset to offset. We measured mean EMG amplitude by integrating the full-wave rectified EMG from onset to offset and dividing it by the selected duration (1 s window for standing and burst duration for locomotion).

## Statistical analysis

We performed statistical analyses with IBM SPSS Statistics 20.0. To determine the effect of training on locomotor recovery, we used a one-factor ANOVA on the locomotor scores to determine statistical differences between the three groups six weeks after spinal transection. For the effect of perineal stimulation on cycle and phase durations, the phasing between hindlimbs and muscle activity, we statistically compared values obtained during quiet standing and locomotion without and with perineal stimulation with a Student's paired t-test for pooled data. To determine if speed significantly modulated cycle and phase durations, the phasing between hindlimbs and EMG activity during tied-belt and split-belt locomotion, we performed a one-factor ANOVA for pooled data. The difference

was significant at an alpha level of 0.05. We did not adjust the α level for multiple comparisons (*Rothman, 1990*; *Hurteau and Frigon, 2018*). We followed the recommendation of *Rothman (1990)* for not correcting for multiple comparisons, in contrast to other views (*Greenland and Robins, 1991*; *Poole, 1991*; *Greenland, 2008*). Proponents of adjusting for multiple comparisons agree with Rothman's recommendation when the objective of the analysis is to scientifically report and interpret the data (*Greenland and Robins, 1991*). Implicit in the assumption that correcting for multiple corrections is needed is that associations between variables in the dataset are manifestations of random processes, or chance (*Rothman, 1990*; *Greenland and Robins, 1991*), which does not apply to our study design. Although this assumption may hold true for certain associations, we do not believe it is valid when considering the association between temporal variables or EMG amplitude at different speeds or left–right speed differences. In this study, we believe that it was more important to avoid false negatives (type II errors) than false positives (type I errors).

## Acknowledgements

We thank Alessandro Telonio for helping with some experiments. We thank Philippe Drapeau from the Rossignol and Drew labs for developing the data collection and analysis software. The present work was supported by grants from the Canadian Institutes of Health Research (PJT-156296), the Merck Sharp and Dohme FMSS Funding Program and the Massage Therapy Research Fund.

## Additional information

### Funding

| Funder | Grant reference number | Author |
| --- | --- | --- |
| Canadian Institutes of Health Research | PJT-156296 | Alain Frigon |
| Merck Sharp and Dohme FMSS Funding Program | | Alain Frigon Nathaly Gaudreault |
| Massage Therapy Research Fund | | Nathaly Gaudreault Alain Frigon |

The funders had no role in study design, data collection and interpretation, or the decision to submit the work for publication.

### Author contributions

Jonathan Harnie, Conceptualization, Formal analysis, Investigation, Visualization, Methodology; Adam Doelman, Emmanuelle de Vette, Etienne Desrochers, Investigation; Johannie Audet, Formal analysis; Nathaly Gaudreault, Conceptualization, Funding acquisition; Alain Frigon, Conceptualization, Resources, Formal analysis, Supervision, Funding acquisition, Investigation, Visualization, Methodology, Project administration

### Author ORCIDs

Alain Frigon https://orcid.org/0000-0002-9259-2706

### Ethics

Animal experimentation: All procedures were approved by the Animal Care Committee of the Université de Sherbrooke and were in accordance with policies and directives of the Canadian Council on Animal Care (Protocol 442-18). Twelve adult cats (> 1 year of age at time of experimentation), 5 males and 7 females, weighing between 3.6 kg and 4.7 kg were used in the present study. Our study followed the ARRIVE guidelines for animal studies (Kilkenny et al. 2010).

### Decision letter and Author response

Decision letter https://doi.org/10.7554/eLife.50134.sa1
Author response https://doi.org/10.7554/eLife.50134.sa2

## Additional files

### Supplementary files
- Source data 1. Source data 1.
- Transparent reporting form

### Data availability
Source data for the figures can be found in Source data 1.

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
