## [Decision Letter]

**Acceptance summary:**

Your study addresses important questions with regards to the necessity of task specific training for restoring motor function following spinal cord injury. Your study provides important evidence that sensory feedback and the restoration of neuronal excitability are key elements in recovery, independent of task specific training. This is an interesting and somewhat unexpected finding and will certainly provoke great interest in the field.

**Decision letter after peer review:**

[Editors’ note: the authors were asked to provide a plan for revisions before the editors issued a final decision. What follows is the editors’ letter requesting such plan.]

Thank you for sending your article entitled "Locomotor recovery after spinal cord injury does not require task-specific training" for peer review at *eLife*. Your article is being evaluated Ronald Calabrese as the Senior Editor, a Reviewing Editor, and three reviewers.

Given the list of essential revisions, including new experiments, the editors and reviewers invite you to respond within the next two weeks with an action plan and timetable for the completion of the additional work. We plan to share your responses with the reviewers and then issue a binding recommendation.

Summary:

It has been demonstrated that locomotion recovers by various means after complete transection in a number of animal models of spinal cord injury due to the presence of a hypothesized network of neurons in the lumbar spinal cord (Central Pattern Generator). It is often assumed that this recovery requires both task-specific training and sensory input to the locomotor neural network. However, few have directly tested if task-specific training is necessary for locomotor recovery, despite this practice now being commonly utilized in clinical trials and physical therapy clinics. The authors hypothesize that recovery of hindlimb locomotion after complete transection does not require task-specific training and, in fact, the field seems to suffer from a lack of appropriate controls or proper analysis of results. In order to test this, the authors spinally transected 12 cats and compared locomotor training (treadmill) vs. manual therapy vs. no training at all (to determine if locomotion recovers spontaneously). Based on their experiments, the authors conclude, after testing for recovery of weight bearing and locomotion (both with EMG of relevant musculature and behavioral scoring), that hindlimb locomotor recovery does not require task-specific training and that weight-bearing recovers spontaneously. This is a provocative finding and challenges the status quo.

Essential revisions:

1) Histology is mentioned in the Materials and methods section but in the Results section it is stated that the data are not shown. The authors should add some qualitative or quantitative data.

2) The results and figures are sometimes difficult to understand. For example, individual cats are mentioned instead of experimental groups and the figures show groups that are not defined in the figure. It would be helpful to define groups on figures with a color code.

3) Since a main point of the manuscript is that task specific and non-task specific training are equally effective at promoting recovery, the analysis of the locomotor data is surprisingly incomplete. A lot of effort was spent on collecting EMG and kinematic recordings (evident given the overwhelming number of recordings and stick figures shown in the paper), a detailed analysis and comparisons between groups are missing (e.g., burst duration, coactivation, left-right coupling, weight support during stepping, stance phase duration etc.). This makes it difficult for the reader to judge if the locomotor pattern does recover similarly between the groups. This could be easily addressed considering the large amount of data collected.

4) Every evaluation is performed with and without perineal stimulation. These data represent a large portion of the figures. Please better justify why the authors performed these two sets of approaches. Also, although, perineal stimulation is a standard in this field, a clear definition of what this means should be included, perhaps even at the start of the results where it is first used. Moreover, the Materials and methods section does not provide details about the protocol for how and when this sensory stimulation was performed – please add those details. Is perineal possibly a confounding factor by adding sensory stimulation? This should be further elaborated, since it seems important for the study.

5) Although each group included 4 animals, not all the animals were tested at all time points. For example, in Group 1 only 2 animals were evaluated for weight-bearing. This would benefit from better explanation and bigger group sizes. Indeed, this was a major concern and raised by all reviewers, and thus, we urge the authors to perform a careful power analysis and justify the number of animals used. Along these lines: the fact that 50% of the cats in the task specific training group (considered the gold standard) did not recover, dilutes the conclusions and indicates the need for more animals. Thus, the authors need to provide some evidence that they tested enough animals to arrive at their provocative conclusions.

6) To make the groups clearer, perhaps the authors could clearly define Groups 1, 2 and 3 at the beginning of the Results section and then stick with this designation throughout. Alternatively, the authors could name the groups (e.g. trained, untrained, non-specific) and put these labels in the table as well.

7) If weight-bearing and locomotor recovery is a spontaneous mechanism in adult cats with a complete SCI, how can it be explained that only half of the animals of Group 2 (receiving specific training) were able to recover in both tasks. Do these findings indicate that specific training is even detrimental?

8) Considering that the primary focus of this manuscript is to examine task specific training, it is interesting that locomotion is also tested using a different paradigm than was used for training. More specifically, training was given on a normal treadmill but a new protocol to assess the locomotion (split-belt locomotion) was tested. There are no real data shown for this and it is unclear how these results add to the study. This part could easily be expanded and needs to be carefully addressed.

9) The suggestion is made that improvements in weight-bearing and locomotion are due to sensory feedback. However, the type of sensory inputs activated and their potential roles in promoting recovery are not further explored. In other words, locomotor training (specific training) and massage (non-task specific training) must have a similar effect since both are stimulating sensory fibers, but this doesn't happen. The authors need to discuss why there are two animals in Group 2 that are unable to support weight or walk despite the sensory stimulation.

10) Another important question is why the cats of Group 3 have a similar recovery to Group 1? How does this support the idea that sensory feedback is essential? Did the massage (sensory stimulation) actually contribute to recovery, is it possible that the unstimulated animals stimulate themselves as they move about their home cages, similar to massage? Please clarify.

11) Weight-bearing was evaluated once every week for all the animals, but it is not clear how this testing was performed and quantified. A 15 seconds window was chosen to evaluate the task. This is a very short time window, which is justified to avoid a training effect. However, there is no reference if the final test (week 6) was performed using the same time window, longer, or how it was performed. It is also possible that improvements in weight-bearing and/or locomotion could be related to spasticity. As training was reported to reduce spasticity, could a decline in spasticity in the trained group explain why some animals performed worse?

12) As the study is somewhat controversial, one would like to see the reported results better integrated into the broader literature. This includes data from humans to rodents, where many studies on spontaneous recovery and/or training have been reported. This is also important given the *eLife* is read by a very general readership.

13) Subsection “Weight bearing recovers spontaneously after spinal transection” – This statement is not correct – In Group 2 only half of the animals were able to recover weight-bearing.

14) It could be preferred to show in Figure 2B the EMG of one cats of each group. Otherwise the contribution of this figure to the story is unclear.

15) Figure 2C/Figure 4B, why were 9 cats used to calculate the mean of EMG with and without perineal stimulation? The motivation for this analysis is unclear and would benefit form more explanation (or the separation of groups).

16) For Figure 4A, please be consistent in the way the EMGs are arranged for each animal.

17) Although knowledge of the mechanisms involved in the recovery of locomotion and weight-bearing is important in the cat, the clinical relevance of this study could be discussed more carefully. Considering the current evidence, the proposal that training maybe not be essential and could be replaced by sensory stimulation extends beyond the scope of the study and should be revised. The authors may also consider studying recovery in animals that are more fully deprived of sensory stimuli (i.e., by explicitly limiting the animals' ability to self-administer sensory stimuli while in their home-cage) to better emulate the scenario in humans that are wheelchair bound.

18) Was weight-bearing or gait analysis blinded for the investigators using the scale? Please clarify.

19) The data should be displayed with all data points for transparency especially in Figure 2C and Figure 4B. In addition, the way they are displayed now makes it seem as if an ANOVA is more appropriate than the t-test values shown. Please modify and clarify the use of your statistical method. Also, the authors should consider the effects of multiple comparisons for the effects of perineal stimulation.

20) Were there any trends in the responses between males and females? Given the new NIH mandate, I think it's worth a cursory analysis (though we do understand that these are very small groups).

21) It seems as if kinematic variability could be a further marker of impairment (or, conversely, recovery) after SCI. Have you considered using something like the angular coefficient of correspondence with hip-knee, hip-ankle, and knee-ankle joint space? (For reference, see Sohn et al., 2018). Or perhaps more kinematic analyses like swing duration, stance duration, interlimb coordination, step height, etc? (for reference see Musienko et al., 2011). These analyses could inform the likelihood of weight-bearing ability after injury which the authors state is necessary for recovery.

[Editors' note: further revisions were requested prior to acceptance, as described below.]

Thank you for resubmitting your work entitled "The recovery of standing and locomotion after spinal cord injury does not require task-specific training" for further consideration by *eLife*. Your revised article has been evaluated by Ronald Calabrese (Senior Editor) and a Reviewing Editor.

The manuscript has been improved but there are some remaining issues that need to be addressed before acceptance, as outlined below:

The authors have generally done an excellent job in revising, except for one important point: it remains difficult to determine how many animals in total recovered locomotion without training. A clear statement of this number would be very helpful, with perhaps some notes on how many were completely untrained and how many got manual manipulations as well as a note about slight variations in timings of locomotor testing. This overall statement could go at the beginning of the Discussion section or end of the Results section.

---

## [Author Response]

[Editors' note: the authors’ plan for revisions was approved and the authors made a formal revised submission.]

Essential revisions:1) Histology is mentioned in the Materials and methods section but in the Results section it is stated that the data are not shown. The authors should add some qualitative or quantitative data.

We have added the histological confirmations of the spinal transections for all cats in Figure 1C and some additional texts describing the surgery.

2) The results and figures are sometimes difficult to understand. For example, individual cats are mentioned instead of experimental groups and the figures show groups that are not defined in the figure. It would be helpful to define groups on figures with a color code.

The groups are now written out on each figure (Non-specific, Locomotor-trained and Untrained) and color-coded.

3) Since a main point of the manuscript is that task specific and non-task specific training are equally effective at promoting recovery, the analysis of the locomotor data is surprisingly incomplete. A lot of effort was spent on collecting EMG and kinematic recordings (evident given the overwhelming number of recordings and stick figures shown in the paper), a detailed analysis and comparisons between groups are missing (e.g., burst duration, coactivation, left-right coupling, weight support during stepping, stance phase duration etc.). This makes it difficult for the reader to judge if the locomotor pattern does recover similarly between the groups. This could be easily addressed considering the large amount of data collected.

We added three new figures and additional data to Figure 4. In Figure 4, we show cycle, stance and swing durations and the phasing between hindlimbs at week 6 after spinal transection without and with perineal stimulation. We also show burst durations and mean EMG amplitude of a triceps surae muscle (soleus or lateral gastrocnemius) as well as the sartorius muscle. In Figure 5, we show adjustments in cycle, stance and swing durations and hindlimb phasing to an increase in treadmill speed for individual cats, separated into the three groups, at week 6 after spinal transection. In Figure 6, we show adjustments to treadmill speed for EMG burst durations and mean amplitude. In Figure 7, we show adjustments during split-belt locomotion for temporal parameters (cycle and phase durations and hindlimb phasing) for the slow and fast limbs.

We also added 9 videos to complement several figures. These videos are referenced in the Results section.

4) Every evaluation is performed with and without perineal stimulation. These data represent a large portion of the figures. Please better justify why the authors performed these two sets of approaches. Also, although, perineal stimulation is a standard in this field, a clear definition of what this means should be included, perhaps even at the start of the results where it is first used. Moreover, the Materials and methods section does not provide details about the protocol for how and when this sensory stimulation was performed – please add those details. Is perineal possibly a confounding factor by adding sensory stimulation? This should be further elaborated, since it seems important for the study.

We provide a more detailed description of the perineal stimulation in the revised manuscript. Perineal stimulation is not a confounding factor because we did not use perineal stimulation during the training period. Testing was done with and without it. Importantly, perineal stimulation is a non-task specific sensory stimulation (tonic stimulation of the perineal skin). As such, it does not change our main point that the recovery and standing and hindlimb locomotion does not require task-specific training.

5) Although each group included 4 animals, not all the animals were tested at all time points. For example, in Group 1 only 2 animals were evaluated for weight-bearing. This would benefit from better explanation and bigger group sizes. Indeed, this was a major concern and raised by all reviewers, and thus, we urge the authors to perform a careful power analysis and justify the number of animals used. Along these lines: the fact that 50% of the cats in the task specific training group (considered the gold standard) did not recover, dilutes the conclusions and indicates the need for more animals. Thus, the authors need to provide some evidence that they tested enough animals to arrive at their provocative conclusions.

That not all animals were tested at all time points was discussed in the paper (subsection “Locomotor recovery after spinal cord injury requires sufficient excitability within the spinal sensorimotor circuits”). At first, we did not want any confounding factors in the recovery of hindlimb locomotion. That is why no weekly testing (weight bearing and locomotion) was done in two animals of the Non-specific group (Group 1) and one in the Untrained group (Group 3). However, much to our surprise, when we observed substantial recovery in the animal of the no training group (Group 3), we decided to perform weekly testing in all remaining animals to better characterize the time course of recovery. The fact that two cats in Group 1 and one cat in Group 3 recovered without any type of testing until week 6 only strengthens our argument that locomotor recovery does not require task-specific training and occurs largely spontaneously. We have clarified this point in the revised manuscript.

It is not clear how adding more animals per group would change our main point that locomotor recovery does not require task-specific training and occurs largely spontaneously. We have 4/4 cats in the Non-specific group (Group 1) and 4/4 cats in the Untrained group (Group 3) that recovered standing and hindlimb locomotion. We are not attempting to demonstrate that one type of intervention is more effective. Adding more cats with a complete spinal cord injury, in our mind, would be unethical and against the Reduction Principle of animal research, particularly with a relatively large mammalian model.

With the addition of new figures and videos, we provide additional support for our conclusions.

6) To make the groups clearer, perhaps the authors could clearly define Groups 1, 2 and 3 at the beginning of the Results section and then stick with this designation throughout. Alternatively, the authors could name the groups (e.g. trained, untrained, non-specific) and put these labels in the table as well.

The groups are now written out on each figure (Non-specific, Locomotor-trained and Untrained) and color-coded.

7) If weight-bearing and locomotor recovery is a spontaneous mechanism in adult cats with a complete SCI, how can it be explained that only half of the animals of Group 2 (receiving specific training) were able to recover in both tasks. Do these findings indicate that specific training is even detrimental?

We agree that our results can be interpreted that locomotor training is detrimental, as the only two cats that did not recover hindlimb locomotion were in that group. However, we do not think this is the case and that is why we presented the results for all cats.

The following sentences were added to the Discussion section:

“The results from our Locomotor-trained group, in which two cats did not recover the capacity for standing or hindlimb locomotion after complete SCI, suggest that locomotor training might even be detrimental. […] We think it a coincidence that the two cats that did not recover hindlimb locomotion fell into the Locomotor-trained group, as the other two cats in this group recovered standing and locomotion to the highest level without perineal stimulation.”

8) Considering that the primary focus of this manuscript is to examine task specific training, it is interesting that locomotion is also tested using a different paradigm than was used for training. More specifically, training was given on a normal treadmill but a new protocol to assess the locomotion (split-belt locomotion) was tested. There are no real data shown for this and it is unclear how these results add to the study. This part could easily be expanded and needs to be carefully addressed.

We agree and we expanded the Results section, adding two new sections that show adjustments to different treadmill speeds during tied-belt (Figure 5 and Figure 6) and split-belt (Figure 7) locomotion. We also show videos for these adjustments.

Split-belt locomotion was done to test locomotor recovery in more demanding conditions but also to test the ability to recover a task that was not trained nor tested during the 6 weeks after spinal transection. This strengthens our conclusion that functional recovery does not require task-specific training.

9) The suggestion is made that improvements in weight-bearing and locomotion are due to sensory feedback. However, the type of sensory inputs activated and their potential roles in promoting recovery are not further explored. In other words, locomotor training (specific training) and massage (non-task specific training) must have a similar effect since both are stimulating sensory fibers, but this doesn't happen. The authors need to discuss why there are two animals in Group 2 that are unable to support weight or walk despite the sensory stimulation.

We believe there is some confusion here. We did not state that improvements in weight bearing and locomotion are due to sensory feedback. The observation that all cats in the Untrained group (Group 3) recovered standing and hindlimb locomotion suggests that the recovery of these functions occurred largely spontaneously. We proposed that the recovery was due to the return of neuronal excitability within spinal sensorimotor circuits, which remain largely intact caudal to the SCI, albeit in a non-functional state due to the lack of excitability. We used perineal stimulation to provide a general increase in spinal excitability. The distinction here is that the expression of locomotion and standing is undoubtedly controlled by sensory feedback, as there is no other way to initiate them in the absence of voluntary commands from the brain. However, their recovery are not driven by task-specific sensory feedback. We added several sentences in the revised manuscript to clarify this point.

We also do not know why two animals from the Locomotor-trained group did not recover, although we can speculate. The following sentences were added to the Discussion section:

“We do not know why these two animals did not recover hindlimb locomotion after spinal transection. Although we can only speculate, their inability to perform hip placement for rostral positioning of the paw and their lack of weight bearing suggests failure in spinal circuits that coordinate motor pools and/or integrate sensory feedback from the limbs.”

10) Another important question is why the cats of Group 3 have a similar recovery to Group 1? How does this support the idea that sensory feedback is essential? Did the massage (sensory stimulation) actually contribute to recovery, is it possible that the unstimulated animals stimulate themselves as they move about their home cages, similar to massage? Please clarify.

As stated in the previous comment, our results suggest that the recovery of hindlimb locomotion after complete SCI occurs largely spontaneously, without the need for task-specific sensory feedback. We had stated that most cats display rhythmic muscle spasms and alternating movements of their hindlimbs when in a sitting condition, which could have contributed to locomotor recovery. Here again, however, this form of ‘self-training’ in non-task-specific. We added several sentences in the Discussion section to clarify this point in the revised manuscript.

11) Weight-bearing was evaluated once every week for all the animals, but it is not clear how this testing was performed and quantified. A 15 seconds window was chosen to evaluate the task. This is a very short time window, which is justified to avoid a training effect. However, there is no reference if the final test (week 6) was performed using the same time window, longer, or how it was performed. It is also possible that improvements in weight-bearing and/or locomotion could be related to spasticity. As training was reported to reduce spasticity, could a decline in spasticity in the trained group explain why some animals performed worse?

How the testing was done was described in the Materials and methods section but this has now been added to the Results section.

The following sentences were added at the start of the Results section:

“We evaluated the recovery of standing and hindlimb locomotion each week after spinal transection in all but three cats (Cat 1 and Cat 2 from Group 1 and Cat 12 from Group 3). […] At week 6 after spinal transection, we did not test standing, but we performed a more thorough investigation of locomotor performance, as described below.”

The problem with spasticity is that it is narrowly defined by some as a resistance to stretch due to high muscle tone or broadly defined by others to also include abnormal patterns of muscle activations, clonus and spasms. We did not evaluate spasticity and based on the definition used, the recovery of hindlimb locomotion could parallel an increase or a decrease in spasticity. We prefer to avoid making these parallels and we feel that this should be the subject of another more rigorous study looking at the recovery of standing and/or locomotion with different definitions of spasticity.

12) As the study is somewhat controversial, one would like to see the reported results better integrated into the broader literature. This includes data from humans to rodents, where many studies on spontaneous recovery and/or training have been reported. This is also important given the eLife is read by a very general readership.

At present with have over 80 references with several discussing findings in rodents, cats and humans.

13) Subsection “Weight bearing recovers spontaneously after spinal transection” – This statement is not correct – In group 2 only half of the animals were able to recover weight-bearing.

Statement was changed to:

“The observation that weight bearing during standing recovered in animals of all three groups without and with stimulation of the perineal skin indicates that it occurred spontaneously, without the need for targeted stand training.”

14) It could be preferred to show in Figure 2B the EMG of one cats of each group. Otherwise the contribution of this figure to the story is unclear.

The purpose of Figure 2B is to show the varying effects of perineal stimulation on individual cats, as there is not consistent effect on a per group basis. The effect of perineal stimulation on a group basis is not particularly interesting because animals show similar levels of recovery in the three groups. To demonstrate this, we chose two cats from the Locomotor-trained group that showed very different effects of perineal stimulation and one cat from the Untrained group.

15) Figure 2C/Figure 4B, why were 9 cats used to calculate the mean of EMG with and without perineal stimulation? The motivation for this analysis is unclear and would benefit form more explanation (or the separation of groups).

As stated, the purpose here is to show the effect of perineal stimulation on different muscles on an individual cat basis and not on an experimental group basis, specifically to show inter-animal variability independent of the intervention received. As such, we pooled data across cats. Stand training was only performed in weeks 1-5 after spinal transection and only included 9 cats. The effects of perineal stimulation during locomotion was quantified at week 6 after spinal transection on 10 cats because 2 cats did not recover hindlimb locomotion.

16) For Figure 4A, please be consistent in the way the EMGs are arranged for each animal.

If the reviewer is talking about the additional stance phases in Cat 3 and Cat 9, this is because the EMGs were from two different episodes due to technical limitations of our set-up (each cat was implanted with two connectors that recorded from 16 different muscles each; we can only record from one connector at a time). We also lost some muscles over time. We modified the figure to show EMGs and stance phases obtained from a single connector/episode.

17) Although knowledge of the mechanisms involved in the recovery of locomotion and weight-bearing is important in the cat, the clinical relevance of this study could be discussed more carefully. Considering the current evidence, the proposal that training maybe not be essential and could be replaced by sensory stimulation extends beyond the scope of the study and should be revised. The authors may also consider studying recovery in animals that are more fully deprived of sensory stimuli (i.e., by explicitly limiting the animals' ability to self-administer sensory stimuli while in their home-cage) to better emulate the scenario in humans that are wheelchair bound.

Our main point is that hindlimb locomotion does not require task-specific training and occurs largely spontaneously. We did not propose that sensory stimulation could replace training. We hope that our results will spur additional research on the effects of sensory augmentation or deprivation. We will add some text to clarify this point. However, working with spinal-transected cats is already difficult from a human perspective for the animal health technicians, veterinarians and lab personnel. Depriving the animals of movement would be difficult to justify ethically.

18) Was weight-bearing or gait analysis blinded for the investigators using the scale? Please clarify.

We now state in the Methods that investigators were not blinded during testing. As we show that recovery did not differ between animals of the three groups, this is a minor point.

19) The data should be displayed with all data points for transparency especially in Figure 2C and Figure 4B. In addition, the way they are displayed now makes it seem as if an ANOVA is more appropriate than the t-test values shown. Please modify and clarify the use of your statistical method. Also, the authors should consider the effects of multiple comparisons for the effects of perineal stimulation.

Data for individual cats are now shown when possible. As we are only comparing two groups (perineal versus no perineal), the paired t-test is appropriate. We also describe in the methods why we did not correct for multiple comparisons.

20) Were there any trends in the responses between males and females? Given the new NIH mandate, I think it's worth a cursory analysis (though we do understand that these are very small groups).

The following sentences were added to the Discussion section:

“Another factor to consider is whether the sex of the animals impacted locomotor recovery after SCI. […] Although, the Locomotor-trained group included 3 females and 1 male, the two cats that did not recover hindlimb locomotion included a male and a female cat.”

21) It seems as if kinematic variability could be a further marker of impairment (or, conversely, recovery) after SCI. Have you considered using something like the angular coefficient of correspondence with hip-knee, hip-ankle, and knee-ankle joint space? (For reference, see Sohn et al., 2018). Or perhaps more kinematic analyses like swing duration, stance duration, interlimb coordination, step height, etc? (for reference see Musienko et al., 2011). These analyses could inform the likelihood of weight-bearing ability after injury which the authors state is necessary for recovery.

We added results on four temporal parameters (cycle, stance and swing durations and phasing/interlimb coordination) over a range of speeds or left-right speed differences during tied-belt and split-belt locomotion, respectively. We also added videos showing examples from each group.

[Editors' note: further revisions were requested prior to acceptance, as described below.]

The manuscript has been improved but there are some remaining issues that need to be addressed before acceptance, as outlined below:The authors have generally done an excellent job in revising, except for one important point: it remains difficult to determine how many animals in total recovered locomotion without training. A clear statement of this number would be very helpful, with perhaps some notes on how many were completely untrained and how many got manual manipulations as well as a note about slight variations in timings of locomotor testing. This overall statement could go at the beginning of the Discussion section or end of the Results section.

The following sentences were added to the start of the Discussion section:

Indeed, all cats that did not receive task-specific locomotor training, which includes 4 of 4 cats in Group 1 (Non-specific) that received manual therapy and 4 of 4 cats in Group 3 (Untrained) that received no intervention or training of any kind after spinal transection recovered hindlimb locomotion (Figure 3). Moreover, 3 cats out of 4 in Group 1 attained the highest performance level on our scale without and with perineal stimulation 6 weeks after spinal transection, whereas 2 and 3 cats attained the highest performance level without and with perineal stimulation, respectively, in Group 3. It should be emphasized that three cats, Cat 1 and Cat 2 from Group 1 and Cat 12 from Group 3 were not tested for standing or locomotor performance in the five weeks after spinal transection. At week 6 after spinal transection, on their very first testing day, these three cats performed hindlimb locomotion at the highest level on our scale, which includes full weight bearing and proper digitigrade placement (Figure 3).